# Measuring diameters and velocities of artificial raindrops with a neuromorphic event camera

Kire Micev[a,1], Jan Steiner[a,1], Asude Aydin[b], Jörg Rieckermann[c], Tobi Delbruck[b,*]

[a]*Department of Mechanical and Process Engineering, ETH Zurich, Zurich, Switzerland*

[b]*Institute of Neuroinformatics (INI), University of Zurich and ETH Zurich (UZH-ETH), Zurich, Switzerland*

[c]*EAWAG, Swiss Federal Institute of Aquatic Science and Technology, Dübendorf, Switzerland*

**Abstract**

Hydrometers that measure size and velocity distributions of precipitation are needed for research and corrections of rainfall estimates from weather radars and microwave links. Existing optical disdrometers

measure droplet size distributions, but underestimate small raindrops and are impractical for widespread always-on IoT deployment. We study the feasibility of measuring droplet size and velocity using a neuromorphic event camera. These dynamic vision sensors asynchronously output a sparse stream of pixel brightness changes. Droplets falling through the plane of focus of a steeply down-looking camera create events generated by the motion of the droplet across the field of view. Droplet size and speed

are inferred from the hourglass-shaped stream of events. Using an improved hard disk arm actuator to reliably generate artificial raindrops with a range of small sizes, our experiments show maximum errors of 7% (mean absolute percentage error) for droplet sizes from 0.3 to 2.5 mm and speeds from 1.3 m/s to 8.0 m/s. Measurements with the same setup from a commercial PARSIVEL disdrometer show similar results. Both devices slightly overestimate the small droplet volume with a volume overestimate of 25%

from the event camera measurements and 50% from the PARSIVEL instrument. Each droplet requires processing of 5k to 50k brightness change events, potentially enabling low-power always-on disdrometers that consume power proportional to the rainfall rate. Data and code are available at the paper website https://sites.google.com/view/dvs-disdrometer/home.

## 1. Introduction

There are increasing numbers of optical disdrometers that measure the diameter and speed of hydrometeors at ground level (X. Liu, Gao, and L. Liu 2013; Johannsen et al. 2020; Löffler-Mang and Jürg Joss 2000; Bailey n.d.; Singh et al. 2021; Rees et al. 2021). Their Drop Size Distribution (**DSD**) measurements are useful for predicting soil erosion and can be combined with weather radars or microwave links to predict a DSD over a larger area (Kruger and Krajewski 2002; Špačková et al. 2021). Laser

disdrometers are commonly installed at airports and wind farms for automated assessment of ground level precipitation. Optical disdrometers measure size and speed of hydrometeors either by direct observation or occulusion of laser sheets, making them potentially easier to calibrate and maintain in the field compared with popular acoustic (Joss and Waldvogel 1967; Joss and Waldvogel 1977; Distromet Ltd. Marc Weibel n.d.) and innovative evaporation-based disdrometers (Singh et al. 2021). The State of the

Art (**SoA**) scientific video disdrometer is the 2-Dimensional Video Disdrometer (**2DVD**) first described by Kruger and Krajewski (2002)[2]. However, 2DVD and competing Particle Size Velocity (**PARSIVEL**) laser-sheet disdrometers have been reported to underestimate total rainfall volume and drift over time resulting in unpractical long-term deployment (Johannsen et al. 2020; Jaffrain and Berne 2011; Upton and Brawn 2008). Different types of disdrometers have been shown to produce measured DSDs that

differ dramatically for small droplets (Johannsen et al. 2020; Cao et al. 2008). They consume significant static power, making it difficult to deploy them in solar-powered weather monitoring, where brownouts can occur in dark weather conditions (Špačková et al. 2021). An event camera disdrometer could provide a field device that would have simple optical and lighting requirements and be accurate over a wide

---

*Corresponding author

  *Email address:* tobi@ini.uzh.ch (Tobi Delbruck)

[1] These authors contributed equally.

[2] See also www.distrometer.at

range of precipitation sizes to enable autonomous continuous DSD measurements that use less power
when there are fewer droplets to measure.

This paper studies the feasibility of using a novel droplet-driven sampling approach based on analyzing
the brightness change events produced by a Dynamic Vision Sensor (DVS) event camera. Such an event
camera does not capture stroboscopic images using a shutter as a conventional camera. Instead, each
pixel reports asynchronous changes in brightness as they occur, and stays silent otherwise (Fig. 1A,
Sect. 2.2) (Lichtsteiner, Posch, and Delbruck 2008; Gallego et al. 2020). They have been successfully
used in many high speed robotics and machine vision applications (Gallego et al. 2020), but not yet in
environmental or atmospheric monitoring.

Our main contributions are:

1. We propose a novel optical disdrometer method that exploits the activity-driven output and high
time resolution of DVS brightness change events to efficiently measure individual droplet size and
        speed using the shallow Depth of Field (DoF) of a fast lens to localize individual droplets in 3d
        space.

2. We generate high-quality ground truth data for small droplets by modifying the Hard Disk Droplet
   Generator (HDDG) from Kosch and Ashgriz (2015) and report how to reproduce this HDDG.

3. We report the first measurements of droplet size and speed with our proposed Dynamic Vision
        Sensor Disdrometer (DVSD) and show that the DSD satisfactorily aligns with the ground truth
        data with at most a mean absolute percentage error of 7%. We show that the DVSD measures small
        droplets with equivalent precision and smaller bias than a commercial PARSIVEL disdrometer.

## 2. Materials and Methods

*2.1. Overview of DVSD Method*

Fig. 1 provides an overview of our proposed DVSD method, which is detailed further in later sections.
The DVS camera (Fig. 1A, Sect. 2.2) asynchronously reports brightness change events as the droplets
pass through a thin DoF at the Plane of Focus Rectangle (PoFR) from a lens that looks down on the
rainfall from a steep angle (Fig. 1B). The droplets are illuminated from behind with a ring of LEDs to
produce bright droplet edges (Appendix F). Each droplet produces about 10k DVS brightness change
events (Fig. 1C). By a simple analysis of this spatiotemporal cluster of events, the DVSD method can
measure both the size and the speed of the droplet (Sect. 2.5.3). We developed a modified Hard Disk
Droplet Generator (HDDG) to generate small droplets (Fig. 1D, Sect. 2.3.2) and used an Intravenous
Dripper Droplet Generator (IVDG) for large droplets (Sect. 2.3.1) . Fig. 1C shows an illuminated
falling water droplet recorded with the DVS camera. Our method consists of two key principles. First,
we aim the camera downward at a steep angle, with an angle $\alpha$ from the vertical (Fig. 1B: left). Second,
the diameters of the droplets crossing the shallow DoF at the Plane of Focus (PoF) are measured
unambiguously, *i.e.*, since the PoF is located at a fixed working distance from the lens, we can infer
the 3d position of the droplet, and hence disambiguate the absolute size from the image size. Droplets
passing through the camera's PoFR come into focus, showing a high contrast and creating an hourglass-
shaped cluster of events, whereas droplets outside the PoFR do not come into focus or create an hourglass
cluster. Droplet velocity is measured by using short accumulation time at two points in time (Fig. 1C,
Sect. 2.5.3). Accumulating the events belonging to one droplet that crosses the PoFR (marked by *
in figure) produces an hourglass shape (Fig. 1C: accumulated hourglass) where the ideal moment for a
droplet diameter measurement (Sect. 2.5.3) is at the waist of this hourglass. The hourglass should be
as concave as possible to facilitate the detection of the waist. Using a fast lens with a small aperture
ratio $f$ number produces a shallow DoF, increasing the amount of blur of the droplets that are out of
focus. Fig. 1B also illustrates how a droplet that crosses the field of view (FoV) but past the PoFR
(marked by # in figure) creates an accumulated image that starts out blurry and becomes increasingly
blurry until it leaves the FoV; similarly (but not illustrated), a droplet that crosses the FoV in front of
the PoFR creates an accumulated event image that starts out blurry and becomes increasingly sharper
until it leaves the FoV. Only droplets that cross the PoFR create hourglasses.

*2.2. Dynamic vision sensor event camera*

The DVSD uses a DVS event camera (Lichtsteiner, Posch, and Delbruck 2008), specifically a DAVIS346[3]
(Taverni et al. 2018). In each pixel (Fig. 2), the logarithmic photoreceptor (**A**) drives a change detec-
tor (**B**) that generates the ON and OFF events (**D**). Pixel photoreceptors continuously transduce the

---

[3]https://shop.inivation.com/collections/davis346

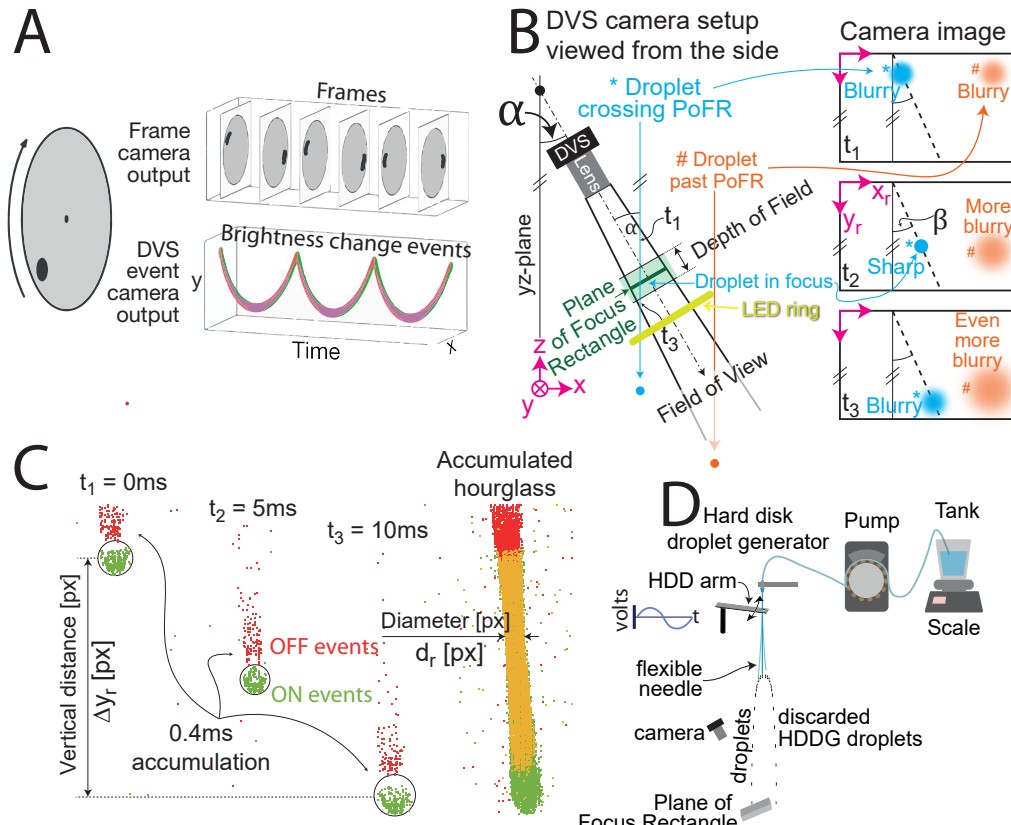

**Figure 1: Summary of Dynamic Vision Sensor Disdrometer (DVSD) methods. A**: Comparison between a conventional frame camera and a DVS capturing a rotating disk with a black dot. The frame camera outputs frames with finite exposure duration at discrete time intervals, whereas the event camera continuously outputs brightness change events, which results in a helix of discrete events in the space time plot (green: increase in brightness, red: decrease in brightness) (Lichtsteiner, Posch, and Delbruck 2008; Gallego et al. 2020) (Sect. 2.2). **B**: Side view of the DVS camera setup in experiments and three illustrations of DVS recordings. The cyan droplet (*) crosses the field of view (**FoV**) and falls through the Plane of Focus Rectangle (**PoFR**), which is tilted at a small angle $\alpha$ from the vertical $yz$-plane. The corresponding camera image is illustrated on the right side at three different times: the cyan droplet (*) enters the FoV; it starts out blurry, comes into focus, and then grows blurry again as it exits the FoV. By contrast, the orange droplet (#) crosses the FoV *behind* the PoFR; it never comes into focus and only grows increasingly blurry. $\beta$ is the angle of the droplet from the vertical $y_r$-axis seen on the recording, caused by droplet velocity component in the $yz$-plane (out of the page). **C**: Sample DVS recording of a droplet crossing the PoFR, which is demonstrated in three frames with 5 ms time differences between each frame. Each of the three DVS frames in this sample is an accumulation of 0.4 ms of events. Green points correspond to ON events, red points show OFF, and yellow points show overlapping of ON and OFF events. The rightmost frame shows all accumulated events over 10 ms. Each droplet creates 10k to 50k events, depending on its size. The diameter $d_r$ of the droplet is measured at the waist of the hourglass when the droplet is in focus, as illustrated on the right. We estimate the falling speed $v_r$ by measuring the focal plane speed of the droplet. Eqs. (1) and (2) provide the physical droplet diameter and speed. (Sect. 2.5.3) **D**: Hard Disk Droplet Generator (**HDDG**) modified from Kosch and Ashgriz (2015). The droplet generator uses a hard disk actuator to oscillate a flexible needle with a constant flow rate of water fed into the needle from a pump. The water tank is placed on top of a scale to calculate the flow rate. Within a $4\times$ range of oscillation frequencies, a droplet is released at each end of the oscillation by large acceleration forces acting on the flexible needle. The diameter of droplets released from the needle is adjusted by the oscillation frequency of the needle. We generated the large 2.5 mm droplets falling 10 m through a circular staircase well with an intravenous (**IV**) dripper.

photocurrent $I$ produced by the photodiode (**PD**) to a logarithmic voltage $V_p$, resulting in a dynamic range of more than 120 dB. This logarithmic voltage (called *brightness* here) is buffered by a unity-gain source follower to the voltage $V_{sf}$, which is stored in a capacitor $C_{DVS}$ inside individual pixels, where it is

continuously compared to the new input. If the change $V_{\mathrm{d}}$ in log intensity exceeds a critical event thresh-
old, an ON or OFF event is generated, representing an increase or decrease of brightness. The event
thresholds $\theta_{\mathrm{on}}$ and $\theta_{\mathrm{off}}$ are nominally identical for the entire array. The time interval between individual
events is inversely proportional to the derivative of the brightness. When an event is generated, the
pixel's location and the sign of the brightness change are immediately transmitted to an arbiter circuit
surrounding the pixel array, then off-chip as a pixel address, and a timestamp is assigned to individual
events. The arbiter circuit then resets the pixel's change detector so that the pixel can generate a new
event. Events can be read out at up to rates of about 10 MHz. The quiescent (noise) event rate is a
few kHz. Events are transmitted from the Dynamic and Active Pixel Vision Sensor (**DAVIS**) chip to
a host computer over Universal Serial Bus (**USB**). The host software records the data and allows play-
back in slow motion. In addition to the DVS circuit the DAVIS346 also has a circuit for conventional
intensity frame recordings called the Active Pixel Sensor (**APS**) circuit (Fig. 2C), which was useful for
lens calibration and focusing.

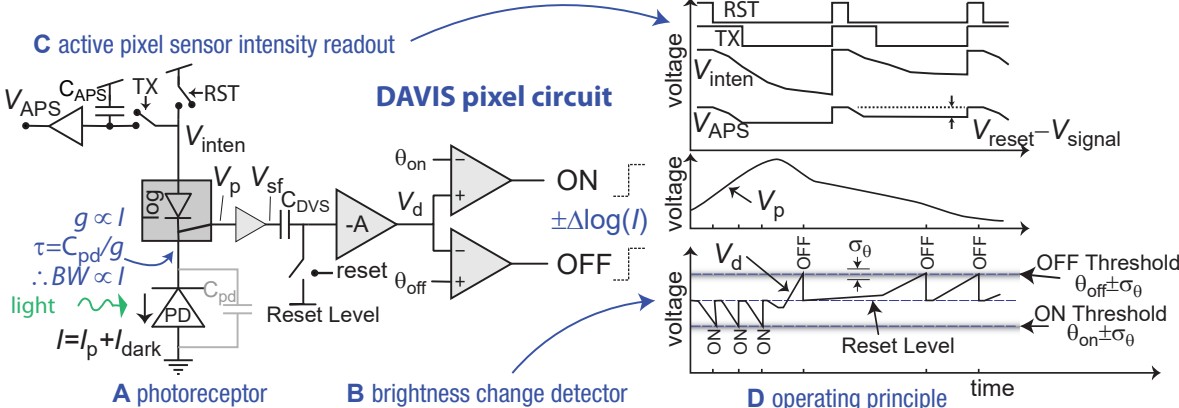

**Figure 2:** DAVIS pixel circuit and operating principle. The sensor generates asynchronous brightness change
ON and OFF DVS events and APS intensity samples, which we do not use for disdrometry, but are helpful
for calibration and focusing. Asynchronous digital circuits place the event $(x, y, p)$ address ($p$ is the event
polarity) onto a shared digital output bus. A USB chip on the camera transmits these events to a host
computer along with the event timestamps $t$. The computer supplies packets of these $(t, x, y, p)$ events with
a maximum latency of 1 ms to user programs.

### 2.3. Artificial droplet creation

Fig. 1D, and Figs. 3 and A1 illustrate our HDDG for creating small droplets. It is based on previous
work by Kosch and Ashgriz (2015), who utilized a computer hard disk arm as an actuator. They used a
high-frequency buzz to create ripples in a steady stream of water emitted by a stiff glass needle, which
would break up into small droplets. Our HDDG uses a flexible plastic needle, which, if properly combined
with a steady stream of water, creates a single droplet at each end of an oscillation, resulting in two
droplet streams, one of which we measured. We used a discarded hard disk drive that we disassembled to
expose the platter head actuator arm. The arm is coupled to the needle by threading the needle through
adhesive tape applied over the hole in the arm allowing the needle to protrude. The arm is actuated with
a home audio power amplifier driven by sinusoidal waveforms generated by an audio wave generating
program where we used coil driver amplitudes from 5 Vpp–20 Vpp and frequencies from 60 Hz–220 Hz.

### 2.3.1. Intravenous droplet generator

Fig. A2 and Fig. 4B shows our setup for creating large droplets with an IVDG assembled from a
standard intravenous dripper obtained from a pharmacy and a needle tip intended for glue dispensers[4].
The diameter of the needle tip (OD 0.4 mm) has only a weak influence on the droplet size ($\approx$2.5 mm),
which is mainly determined by the surface tension of water adhering to the needle; at low flow rates,
when the droplet mass grows large enough, it breaks free from the needle. We adjusted the IV flow to
produce a regular series of droplets at a rate of about 1.6 Hz (Appendix E).

---

[4]Gray color, with ID 0.2 mm, OD 0.4 mm, part VD90.0032 https://www.martin-smt.de

### 2.3.2. Hard disk droplet generator

Fig. 1D and illustrate our HDDG and Figs. 4 and A1 shows the HDDG setup. Our HDDG is based on previous work by Kosch and Ashgriz (2015), who utilized a computer hard disk arm as an actuator. They used a high-frequency buzz to create ripples in a steady stream of water emitted by a stiff glass needle, which would break up into small droplets with about 100 um diameter. Our HDDG uses a soft and flexible plastic needle, which, if properly combined with a controlled stream of water, reliably creates
a single droplet at each end of an oscillation, resulting in two droplet streams, one of which we measured.

*HDDG droplet needle.*  To make the droplet needle, we use a microloader microcapillary tip[5] This soft plastic needle tubing protrudes from its integrated feeder expansion. The peristalitc pump tubing[6] is plugged into this microloader. The needle is threaded through a hole drilled through the hard disk drive (**HDD**) actuator arm so that the needle protrudes from the arm by 2 cm–4 cm, and we can control the
length of the protruding needle to adjust its resonance frequency to match the driving frequency. That way, we can use a smaller driving voltage and current for the HDD driver coils. The needle is fixed to an elevated platform with a conical interference fit between the needle and a black plastic tube glued to that platform (see Fig. A1 C and D).

*Construction of HDDG droplet generator.*  Fig. 3 sketches the HDDG construction. We use an old 250GB
3.5" hard disk drive that we disassembled to expose the platter head actuator arm. The coil driver wires have two functions: first, to power the HDD coils and second, to act like springs to keep the HDD arm close to the middle of the two permanent magnets, which is the best operating point for the arm. To construct the needle driver, we follow these steps:

1. The upper end of the plastic needle is frictionally held in place (see black tube).
2. Two nuts and a bolt (Fig. A1D) adjust the protruding length of the needle to match the resonance frequency of the needle with the HDD frequency to maximize the amplitude of the oscillation and, therefore, the efficiency.
3. The lower end of the plastic needle is guided through a tiny hole in a piece of sticky tape. The tape is attached to the end of the HDD arm.
4. The needle is connected to the water tube from the pump. This connection must be tight to prevent water leakage and to prevent the needle from twisting.
5. Since the needle has some inherent curvature, we twisted it with our fingers until the inherent curvature was perpendicular to the oscillation direction.

*Driving the HDD.*  The HDD arm is actuated with an audio power amplifier driven by sinusoidal wave-
forms generated by an audio wave generating program (www.szynalski.com/tone-generator). The arm is coupled to the needle by threading the needle through one-sided adhesive tape which is applied over the hole in the arm. We used amplitudes from 3 Vpp–10 Vpp and frequencies from 60 Hz–220 Hz. By adjusting the flow rate and oscillation frequency, we can arrive at a combination of settings where nearly on every oscillation, a single droplet is flung from the needle tip at each end of the oscillation. Since the
flow rate and frequency are constant, the droplet sizes are also constant.

### 2.4. Experimental setups for DVSD and PARSIVEL measurements

Fig. 4 illustrate the HDDG and IVDG setups. Figs. A1 and A2 show detailed photos of the setups.
We replicated both sets of experiments with the commercial PARSIVEL instrument listed in Table 4. For the PARSIVEL IVDG experiment, we used a different stairwell that is much more open to wind
currents because the circular one used for the DVSD experiments was not available.
For the DVS measurements we optimized the camera biases for droplet observation by increasing the pixel bandwidth and optimizing the event threshold settings. The DVS are recorded on disk as AEDAT-2.0 files using jAER (https://jaerproject.net).
For the PARSIVEL measurements, we used the default device settings. The PARSIVEL records the
droplet class counts and writes them to summary CSV files.

---

[5] OD 0.3 mm Eppendorf 20 ul microcapillary pipette; Merck Catalog No. 930001007
[6] OD 3.5 mm Tygon® S3™E-3603, Saint-Gobain Performance Plastics; Tygon tubing website

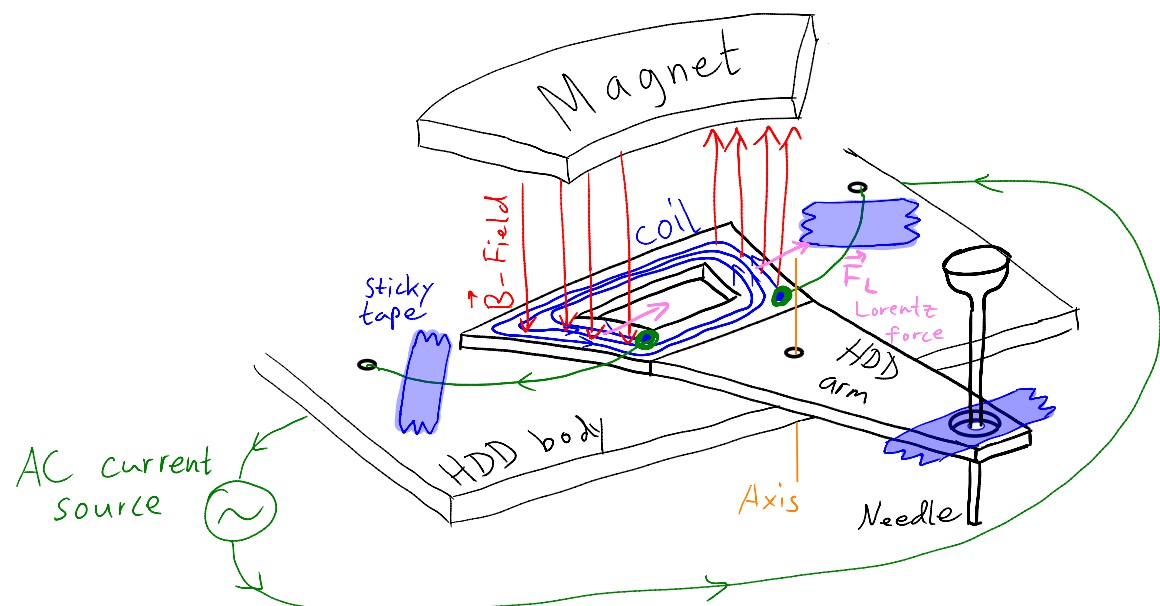

**Figure 3: Sketch of HDDG.** The red B-vector is the induced magnetic field by the coil and alternates its direction from up to down. When it points up, it is attracted to upwards pointing yellow B-vectors from the static magnet. Two sticky tapes and two green wires hold the HDD arm in place in a spring-like manner. This attachment ensures that the coil is centered between the two opposite poles of the metal piece. Without the sticky tapes, we observed that the arm drifts to one side and does not function properly anymore. See Fig. A1 for photos.

### 2.5. Experiments

The objective of the experiments was to measure the diameter and velocity of droplets with a DVS with droplets falling close to their terminal speed, and to compare these measurements with their corresponding ground truth (**GT**) (Sect. 2.5.4), and to PARSIVEL measurements using the same setups (Sect. 3.2). Two experiments were conducted with different setups to optimize droplet creation for two different drop size categories, which are further explained in the following two paragraphs. Both experiments used the same DVS camera and measurement principles (described in Sect. 2.5.3). Lenses for each experiment were chosen to make the droplets create at least 10 pixel diameter at the PoF. Both lenses were set at their minimum focusing distance. The working distance was then reduced further to about 50 cm using lens spacers.

Table 1 compares both experimental setups with further details. Both droplet generation methods are described in Sect. 2.3.

The first experiments, called HDDG experiments, were conducted in a darkroom using an HDDG and a fall height of 2 m. This setup was optimized for droplets with a diameter of 0.3 to 0.6 mm, by using a 300 mm lens with the resulting large magnification $M = 30.7$ px/mm. This led to a sampling area of 0.9cm$^2$. The HDDG produced a localized drop jet, making it fairly easy to hit the sampling area. The height of the droplet fall is enough for the drops to reach more than 99% of the terminal velocity according to Appendix H. We used a 40 W Light Emitting Diode (**LED**) ring purchased from a home supply store as a light source pointing upward and inward to achieve a high contrast between the bright drop edges and the dark background (Fig. A1A and Appendix F).

The second experiment, called the IVDG experiment, was carried out in the vertical tunnel of a spiral staircase using a IVDG as a droplet generator and a fall height of more than 10 m. With a smaller magnification of 7 px/mm (4.4× smaller than for the HDDG experiments), the magnification was reduced for droplets with a diameter of approximately 2.5 mm, while maintaining a relative precision similar to that of the HDDG setup. The main reason for this reduction in magnification was to more easily capture the droplets, which had a huge scatter compared to the HDDG scatter. With the IVDG, it was only possible to create a single droplet size because the droplet size is determined by the weight that breaks the surface tension with the needle. The higher magnification led to a larger sampling area of 4cm$^2$, increasing the chance of capturing the larger drops, which have a much greater spread from the higher

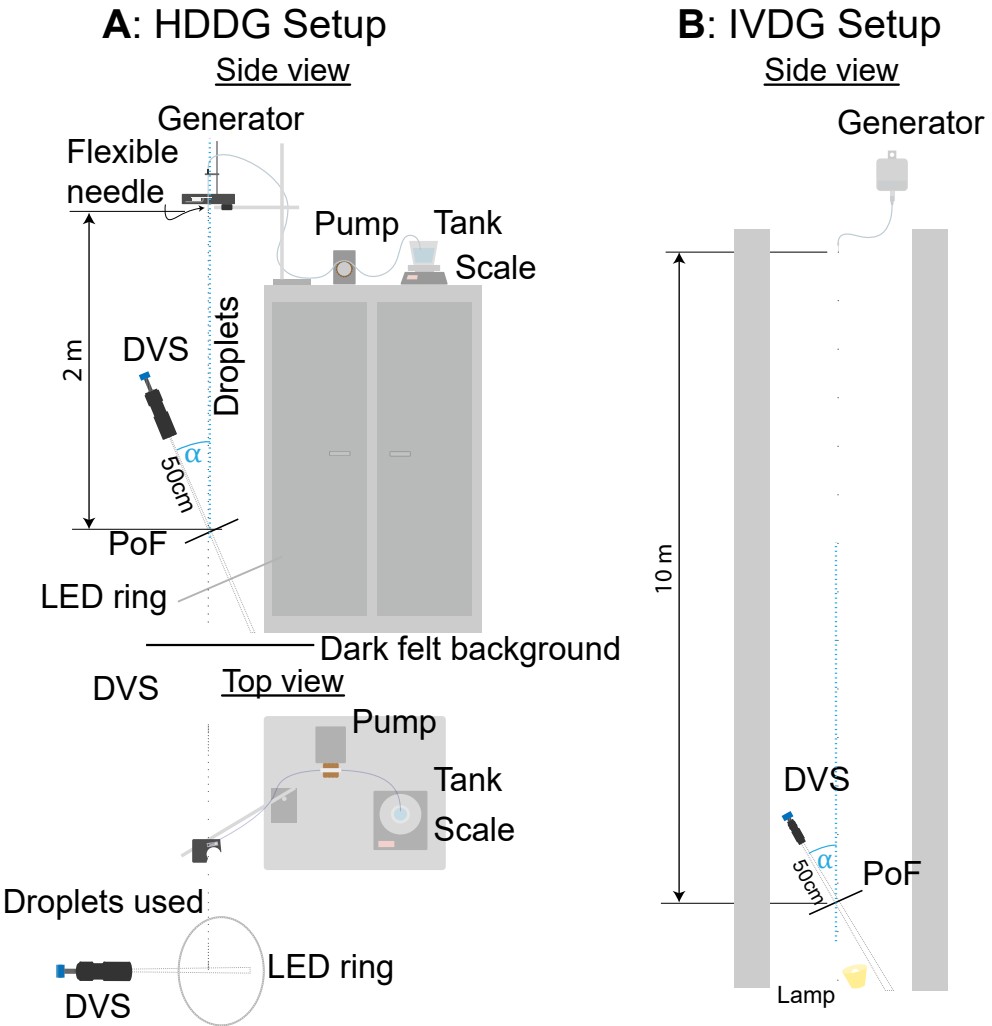

**Figure 4: Illustration of experimental setups. A:** HDDG setup is illustrated from a side view and a from a top view perspective. **B:** IVDG setup is illustrated from a side view perspective.

fall height needed to achieve a final speed close to the terminal speed. The height of the fall is sufficient for the drops to reach more than 97% of the terminal velocity according to Appendix H. The working space of the experiment IVDG was more limited (Fig. A2B), so we used a single 5 W LED reading lamp to illuminate the drops from behind, which refracted the light towards the middle of the drop, leaving the edges dark. Despite this non-optimal lighting, we could still determine the width and velocity of the droplets from the recordings.

A key factor was the adjustable angle of the camera from the vertical $\alpha$ (see Fig. 1B: left). The smaller is $\alpha$, the more accurately the diameter can be measured and the larger is the horizontal sampling area, which leads to a faster estimate of DSD. Larger $\alpha$'s allow more precise velocity measurements, but smaller sampling area. The sampling area is the total area of the PoF inside the FoV multiplied by $\cos \alpha$. According to our findings, 20° to 30° was optimal for $\alpha$.

Eq. (1) of Sect. 2.5.3 was used for the diameter calculation. To compute the velocity, we used the non-simplified formula on the left side of Eq. (2) for the HDDG experiment, and used the simplified formula on the right side of Eq. (2) for the IVDG experiment due to the negligible horizontal velocity component on the recording (see Fig. 1B: right). Appendix G describes how we measured the parameters.

*2.5.1. Data selection*

Each recording session collected data for a single droplet size. From the recording, we manually selected droplets that crossed the image and created a distinct hourglass shape. These droplets cross

the PoFR sampling area and thus allow inferring a reliable DSD. Fig. 1 illustrated how this procedure excluded droplets that did not pass through the PoF.

| Setup parameters | Experiment | |
| --- | --- | --- |
| | HDDG | IVDG |
| Drop diameter | 0.3–0.6 mm | 2.5 mm |
| Drop velocity | 1.4–2.2 m/s | 7.4 m/s |
| Fall height | 2 m | 10 m |
| Lighting (Appendix F) | 40 W LED ring | 5 W LED bulb |
| Total luminance | 4500 lm | 470 lm |
| Location | darkroom | spiral staircase |
| Lens | TAIR-3 (Russian sniper rifle) | Edmund manual zoom |
| Listed focal length $f$ | 300 mm | 75 mm |
| Aperture ratio $N$ | f/4.5 | f/1.2 |
| Lens Spacer | M42-C (16 mm long) | - |
| # 5 mm C-CS lens spacers | 23 | 2 |
| Working distance $w$ | 50 cm | 50 cm |
| Sampling area (Appendix G.1) | 11×8 mm$^2$ | 49×32 mm$^2$ |
| Camera angle $\alpha$ | 22° | 29.5° |
| Magnification $M$ (Appendix G) | 30.7 px/mm | 7.0 px/mm |
| DoF (Appendix G.2) | 0.46 mm | 1.97 mm |

**Table 1:** Detailed setup parameters for the HDDG and IVDG experiment. See also Table A1.

*2.5.2. Measuring image plane droplet diameter and velocity from DVS event recording*

The image plane droplet diameter $d_r$ and velocity $v_r$ are measured from the DVS recording with jAER (https://jaerproject.net). The *Speedometer* [7] plugin filter allows for convenient measurement of the diameter and velocity by outputting the velocity seen on the recording $v_r$ [kpx/s], horizontal displacement $\Delta x_r$ [px] and vertical displacement $\Delta y_r$ [px].

Fig. 5A shows how hourglasses appear on a DVS recording after accumulating events from two HDDG droplets that passed through the FoV. Only the droplet on the right side is analyzed. The width at the center of the hourglass indicates the diameter of that droplet when in focus (Fig. 1C on the right). This width $d_r$ [px] is measured with jAER's *Speedometer* plugin using two mouse clicks. The right point at the thinnest width of the *hourglass* is selected first (see Fig. 5A). Next the left point on the thinnest

width of hourglass is selected. *Speedometer* displays the diameter of the recording $d_r = 15$ px.

Fig. 5B&C shows the droplet image plane velocity measurement. For this measurement, the event integration time is decreased to about 0.5 ms per frame. Before the drop reaches the PoF (smallest diameter), the playback is paused and the midpoint of the drop is selected by a mouse click (Fig. 5B). After the drop has passed the PoF (Fig. 5C), the playback is again paused and the midpoint of the

droplet is again selected. The *Speedometer* outputs the velocity of the droplet in the image plane as $v_r = 20.8$ kpx/s.

*2.5.3. Computing physical droplet diameter and velocity from DVS image plane measurement*

Given the image plane droplet diameter $d_r$ (Sect. 2.5.2), the physical droplet diameter $d$ is calculated from Eq. (1), where $M$ is the calibrated camera magnification (Appendix G):

$$d \text{ [mm]} = \frac{d_r \text{ [px]}}{M \text{ [px/mm]}}. \tag{1}$$

For computing droplet velocity measurement $v$ from $v_r$, it is useful to use a lens with a long focal length, resulting in a small Angle of View (AoV) $\theta_i$, so that the magnification $M$ at the PoF can be assumed to be constant. To further mitigate this effect, it is important to measure the velocity as close as possible to the center of the hourglass. The droplet fall velocity $v$ is calculated from Eq. (2), where

---
[7]Speedometer class. Speedometer usage.

**Figure 5: A: Diameter measurement.** Measurement of the diameter from the DVS recording using jAER. Two droplets passed during the accumulation time. After marking the left and right sides of the hourglass waist, *Speedometer* displays the diameter $d_r$ components *dx* and *dy* [px]. **B and C: Velocity measurement.** Measurement of the velocity from the DVS recording using jAER. **B**: Mark IN point first on the mid-point of the right drop before the drop passes the PoF. **C**: Mark the OUT point at the midpoint of the drop after the right drop passes the PoF, then *Speedometer* displays the velocity $v_r$ [kpx/s]. This is a sample recording of an HDDG drop with a droplet creation frequency $f$ of 100 Hz (2×50 Hz, two-sided).

$\Delta x_r$ and $\Delta y_r$ are the droplet image displacement and $\Delta t$ is the time interval of the displacement:

$$v \text{ [m/s]} = \frac{\sqrt{(\Delta x_r \text{ [px]})^2 + \left(\frac{\Delta y_r \text{ [px]}}{\sin \alpha}\right)^2}}{10^3 \cdot \Delta t \text{ [s]} \cdot M \text{ [px/mm]}} \overset{(*)}{\approx} \frac{\Delta y_r \text{ [px]}}{10^3 \cdot \sin(\alpha) \cdot \Delta t \text{ [s]} \cdot M \text{ [px/mm]}} = \frac{v_r \text{ [kpx/s]}}{\sin(\alpha) \cdot M \text{ [px/mm]}} \quad (2)$$

Eq. (2) is simplified to the second form when $\Delta x_r/\Delta y_r \ll 1$ and therefore (*) $\Delta y_r \gg \Delta x_r$, i.e. when $\beta$ in Fig. 1B is small. This simplified formula can only be applied if the drop falls vertically for a correct velocity calculation, otherwise, $\alpha$, which is used for the velocity simulation, would not be constant anymore. Fig. 1C shows an abstract illustration of a measurement of diameter and velocity from a DVS recording, where the black circles show where the actual droplets are. The circles enclose the bottom edge of the ON events (green points) and OFF events (red points). The tracking points of the droplets that we used for the DVS velocity estimates are the centers of the black circles. Sect. 2.5.2 shows examples of our actual image plane measurements of diameter and velocity. For the droplet in Fig. 5 using the $M$ and $\alpha$ HDDG values in Table 1, Eq. (1) evaluates to $d = 15\text{px}/30.7$ px/mm $= 0.48$ mm and Eq. (2) evaluates to $v = 1.8$ m/s.

*2.5.4. Ground truth droplet size and velocity determination*

Our GT droplet size measurements (not to be confused with ground level) provide the reference data for droplet size and speed that we compare with the DVSD and PARSIVEL measurement results.

The HDDG (Sect. 2.3.2) needle oscillation frequency is adjusted with a function generator and resulted in droplet diameters between 0.3 and 0.6 mm, which in our case corresponded to a droplet creation frequency between 60 and 220 Hz and flow rate $5.19 \times 10^{-3}$ g/s. At An ISMATEC REGLO ISM597A Digital peristaltic pump[8] supplied our HDDG with water at constant flow rate from a tank placed on top of a KERN 440-35A weighing scale[9] with 0.1 mg precision. The scale measured the water mass over time to determine the flow rate. By assuming spherical water drops, their diameter is inferred from their mass, which increases with flow rate but decreases with drop creation frequency. The drop creation frequency describes how many drops per second are produced. According to a water droplet model proposed by Beard and Chuang (1987), a spherical assumption is very accurate for submillimeter drops.

The volumetric flow rate $\dot{V}$ is calculated as Eq. (3):

$$\dot{V} = \frac{\Delta m}{\rho \Delta t} \quad (3)$$

where $\rho = 0.997$g/cm$^3$ is the density of water, $\Delta m$ is the change in mass of the water tank (HDDG) or collection cup (IVDG), over a certain period of time $\Delta t$. With the flow rate $\dot{V}$ and the assumption of a

---

[8] https://us.vwr.com/store/item/NA5192025/masterflex-ismatec-reglo-digital-variable-speed-multichannel-peristaltic-pumps-
[9] https://www.kern.swiss/de/waagen/laborwaagen/praezisionswaagen/kern-440-35a

**305**  sphere, the diameter is calculated as Eq. (4):

$$d = \left(\frac{6\dot{V}}{\pi f}\right)^{1/3} \tag{4}$$

where $f$ is the drop creation frequency.

We used the IVDG of Sect. 2.3.1 to create larger droplets. To determine the mass of a single droplet, many droplets were counted and collected in a reservoir positioned on the scale while the total mass $m_{\text{tot}}$ and the number of drops $N$ were recorded. The volume of a single drop is calculated as

$$V_{\text{drop}} = \frac{m_{\text{tot}}}{\rho N} \tag{5}$$

**310**  The diameter can then be calculated as

$$d = \left(\frac{6V_{\text{drop}}}{\pi}\right)^{1/3} \tag{6}$$

where the droplet is assumed to be a sphere. The diameters calculated from Eq. (4) and Eq. (6) are used as GT values to compare with the diameter measurements of DVS.

A separate measurement (Appendix E) shows that the IVDG droplet size distribution varies less than $\pm 1\%$ over short time scales and less than 3% over the 20m measurement duration. For these droplets, **315** the model by Beard and Chuang (1987) predicts a slight average drop deformation due to drag. However, this average deformation is 0.8% in the case of 2.50 mm droplets, making them appear to be 2.52 mm when viewed from roughly 30°. The average deformation thus introduced only a very small systematic DVS error. A random error was additionally introduced from the vibration of the 2.5 mm droplets.

We computed the GT velocity from the GT diameter using a droplet velocity simulation (Appendix H) **320** for the small HDDG droplets. For the large IVDG droplets, we used the measured velocity from Gunn and Kinzer (1949).

### 2.5.5. Error analysis

In both experiments, two aspects were considered to determine the propagation of the error of the measurements.

**325**  The first aspect is the combined measurement uncertainty of the DVSD and GT values which factors in all uncertainties. For example, one measurement uncertainty of the DVSD comes from the limited sensor resolution of the DVSD. Another one comes from the uncertainty of $\alpha$. Together with other uncertainties, we could then calculate the combined standard uncertainty of the DVSD velocity and diameter measurements. The combined GT uncertainty consists of the uncertainty in height, droplet **330** mass, drop creation frequency, etc. According to JCGM (2008), in the case of uncorrelated input quantities $x_i$, the combined standard uncertainty of a function $u(f)$ can be described as:

$$u(f) = \sqrt{\sum_{i=1}^{N} \left[\frac{\partial f}{\partial x_i} u(x_i)\right]^2} = \sqrt{\left[\frac{\partial f}{\partial x_1} u(x_1)\right]^2 + \left[\frac{\partial f}{\partial x_2} u(x_2)\right]^2 + \left[\frac{\partial f}{\partial x_3} u(x_3)\right]^2 + \dots} \tag{7}$$

where $u(x_i)$ is the standard uncertainty of the input quantity $x_i$. The uncertainty is either derived with a *Type A evaluation*, where the standard uncertainty is evaluated with the experimental standard deviation from repeated observations, or with a *Type B evaluation*, where the estimated uncertainty **335** is evaluated using our judgement of uncertainty. We used a *Type B evaluation*, either by using the manufacturer's instrument specifications or by a conservative estimate of the measured uncertainty. A linear approximation of the function is used for each input quantity. A spreadsheet in our online results folder[10] reports the final uncertainty values computed by our Matlab scripts.

The second aspect is accuracy, *i.e.*, comparison of the measured values from the DVS with the GT

---

[10]Results computations; see 00 README.txt

**340**  values. This is done by calculating the Mean Absolute Percentage Error (MAPE):

$$\text{MAPE} = \frac{100\%}{n} \sum_{i=1}^{n} \left| \frac{GT_i - ME_i}{GT_i} \right| \tag{8}$$

where $GT_i$ is the GT value and $ME_i$ is the measured value.

## 3. Results

We conducted two series of DVSD experiments, one with the HDDG and one with the IVDG. We used different lenses to make it easier to capture droplets crossing the PoF. The droplets created by
**345**  the HDDG ranged from 0.3 mm to 0.6 mm (10 to 20 pixel diameter on the image), while the droplets created by the IVDG were 2.5 mm (17-18 pixel diameter). In both experiments, the height of the fall was sufficient for the droplets to reach within 97% of the terminal speed (Appendix H). Because the DVSD sampling area is small, these experiments required patience to collect sufficient data, i.e., such that the recorded droplet showed an hourglass shape in the accumulated playback of the DVS recording. In total
**350**  we collected $N = 45$ droplets with the HDDG setup and $N = 27$ droplets with the IVDG setup.

To allow comparison, we repeated the experiments with the commercial PARSIVEL laser sheet disdrometer in Table 4. Its larger sampling area (Table 4) allowed us to collect thousands of droplets.

### 3.1. Results from DVSD

Fig. 6 compares the DVSD measurement results performed with the DVS (Sect. 2.5.3) to GT estimates
**355**  of droplet size and speed (Sect. 2.5.4). The HDDG droplets (green data points) are magnified for better visibility, and the IVDG droplets (purple data points) have a purple histogram beside them, indicating the number of measurement results that overlap. The quantization of the data arises from the quantized droplet size generation and the pixel discretization. Horizontal quantization is caused by the quantized HDDG droplet creation frequencies, which control the diameters of the droplets. Vertical quantization
**360**  is caused by the low pixel count of the diameter of the droplets in the DVS recording. The speed measurements do not have any significant vertical quantization effects, due to large pixel displacements ($\approx 100$ pixels) and the fine DVS event timestamp resolution of 1 µs. The DVSD results are discussed in Sect. 4.1.

**375**  We used MAPE to quantify the discrepancy between the ground truth values and the DVS measurements (Sect. 2.5.5). Table 2 lists the diameter and velocity MAPE for both experiments. In all cases, MAPE remains below 7%, which consists of the DVSD bias to overestimate the droplet diameter.

**Mean absolute percentage error**

| Experiment | Diameter | Velocity |
|------------|----------|----------|
| HDDG       | 6%       | 4%       |
| IVDG       | 7%       | 4%       |

**Table 2:** MAPE of the DVS measurements compared to GT values, for the diameter and velocity measurements from both experiments.

Table 3 lists the estimated combined uncertainties of the measured diameter and velocity using the method explained in Sect. 2.5.5. In some cases, a range of uncertainties is given, which means
**380**  the uncertainty depends on the size of the droplet. The combined uncertainty percentage is largest for the DVS measurement of the smallest droplets created with the HDDG, which had a diameter of 0.35 mm $\pm$ 0.03 mm ($\pm 10\%$). This large uncertainty is caused by the low pixel diameter count ($\approx 10$ pixels).

The uncertainty (*i.e.*, precision) of DVSD diameter and velocity measurements are mainly limited by
**385**  the spatial resolution of the $346 \times 260$-pixel DVS (Table 3: bottom). The GT diameter uncertainty (see Table 3: top) was mostly caused by the measurement uncertainty of the scale and noise in the droplet generation by the HDDG and IVDG. The GT velocity uncertainty arises from neglecting air turbulence, droplet deformation, and inaccurate sphere model values, i.e. GT droplet diameter estimates from mass flow. The GT droplet diameter and velocity are calculated from their mass and from the simulation,
**390**  respectively (see Sect. 2.3 and Appendix H). Therefore, if the diameter is uncertain, it increases velocity uncertainty.

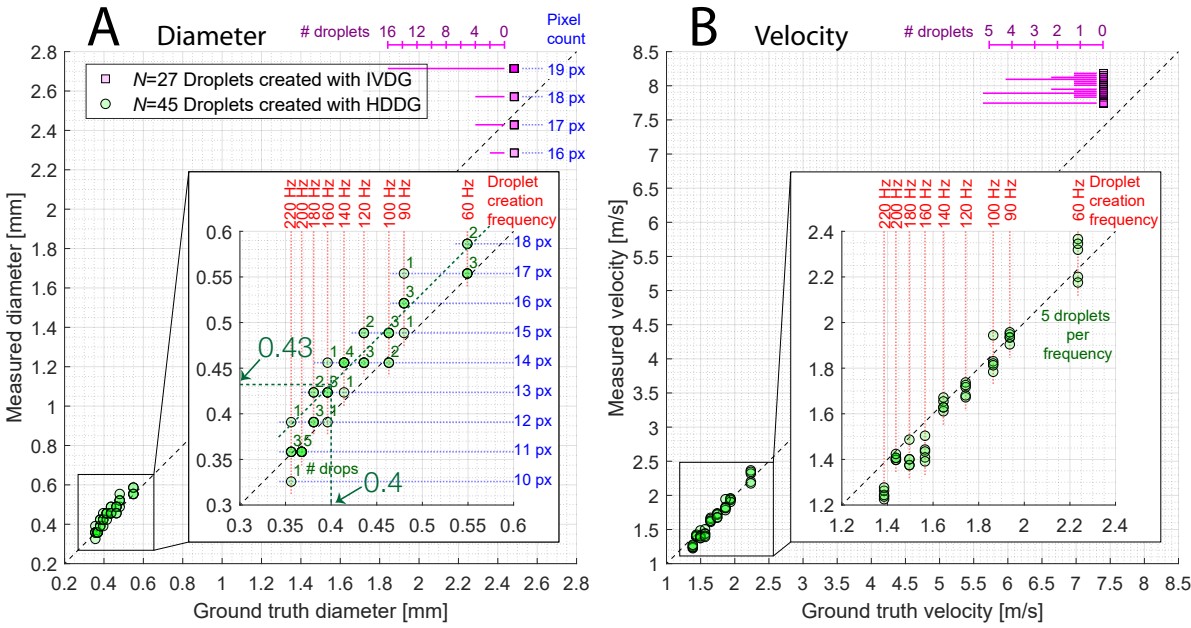

**Figure 6: Main results.** DVSD measurement results of droplets compared to GT values, where **A** shows the diameter and **B** shows the velocity. The dashed black line represents a 45°-line passing through the origin. Both droplet creation methods are included in the plots. The zoomed plots show droplets created with the HDDG for improved visibility. Numbers adjacent to the points on the left zoomed plot indicate droplet overlaps whereas the zoomed plot on the right has 5 droplets per frequency. In **A**, the dashed green lines show a fit to the small droplet data and the measured diameter of 0.43 mm corresponding to the 0.4 mm GT droplets. The IVDG droplets are shown in purple with a histogram for number of overlapping points. Quantization effects caused by the pixel count or frequency are illustrated as a grid pattern in the plots where the effect is significant. (See Sect. 3 for details.)

**Combined uncertainty**

| Method | Experiment | Diameter ±[%] | Diameter ±[mm] | Velocity ±[%] | Velocity ±[m/s] |
|--------|-----------|------|--------|------|--------|
| GT | HDDG | 3 | 0.01–0.02 | 4–7 | 0.1 |
| | IVDG | 2 | 0.05 | 4 | 0.3 |
| DVS | HDDG | 6–10 | 0.03 | 7 | 0.1–0.2 |
| | IVDG | 6–7 | 0.15 | 6 | 0.5 |

**Table 3:** Percentage and absolute combined uncertainty of the DVS measurements and GT values for diameter and velocity.

### 3.2. Results from PARSIVEL disdrometer

Fig. 7 summarizes the results obtained from the commercial PARSIVEL disdrometer in Table 4 using the HDDG and IVDG setups. Detailed histograms of these results are reported in Appendix C. The PARSIVEL results are discussed in Sect. 4.2.

## 4. Discussion

### 4.1. DVSD experimental results

The DVSD results presented in Fig. 6 show excellent linearity over the entire measurement range for both size and speed; the dashed line in each plot has a slope of one and passes through the origin; it lies close to both small and large droplet measurements. Size measurements overestimate droplet diameters by about 8%, and speed measurements slightly underestimate large droplet speeds.

Although the large droplets were generated and measured differently than the small droplets, the two data sets are very consistent (Fig. 6: green and purple data sets). The offset of the data points from the 45°-line (Fig. 6) might be correctable. We believe it most likely arises from the method of manual marking of the accumulated events to determine the hourglass waist width; this method could slightly

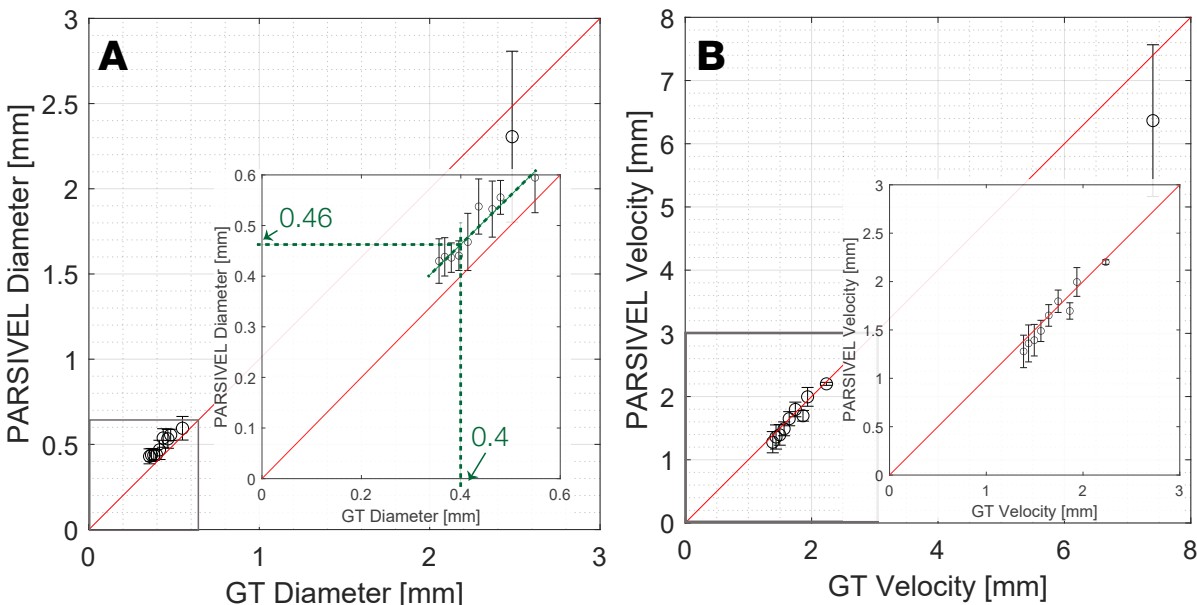

**Figure 7:** Summary of results from PARSIVEL in Table 4. Each plot includes a magnified view of the HDDG small droplet results. The red lines pass through the origin with a slope of one to indicate ideal measurements. **A:** Diameter measurements vs. GT. the green dashed lines show a fit to the small droplet data and the corresponding GT and measured diameter values. **B:** Velocity measurements vs. GT. See Figs. A3 and A4 for detailed distributions.

overestimate the droplet width. The alternative (an inaccurate $M$ estimate) we consider less likely. Our measurements show from the small droplet fit that GT droplets with diameter 0.4 mm are estimated by the DVSD method to be about 0.43 mm, an overestimate of 8% in diameter and 24% in volume. Since the HDDG droplets measure about 10 px in the image plane, a single pixel overestimate of droplet width could explain the discrepancy.

We used a shorter lens for the larger IVDG droplets only to allow us to capture the large droplets more easily, since they scattered much more from random wind currents in the staircase than the small droplets from the HDDG. It is possible to obtain better precision of large droplets by using the same lens, but with the trade-off of longer experiment time since fewer will pass through the PoFR.

For the IVDG experiment with large droplets, the DVSD method overestimated the the 2.5 mm droplets computed GT speed value of 7.4 m/s as about 7.8 m/s, a 5% overestimate. (Appendix H). The droplet velocity distribution appears to have a long tail of droplets with smaller velocities.

*4.2. PARSIVEL experimental results*

In general, the PARSIVEL measurements in Fig. 7, Figs. A3 and A4 agree fairly closely with our GT droplet sizes and velocities and have precision similar to the DVSD, but they consistently show a larger bias to overestimate the small droplet sizes. This size bias is evident in the inset plot of Fig. 7A, where we fitted a line to the PARSIVEL results. The dashed green lines show that GT droplets with diameter 0.4 mm are estimated by the PARSIVEL to be about 0.46 mm, an overestimate of 15% in diameter and 50% in volume. By contrast to the DVSD method which overestimated the 2.5 mm droplet speed by 5%, the PARSIVEL reported speed of only 6.4 m/s (a 16% underestimate). However a caveat to this result is that the PARSIVEL velocity distribution in Fig. A4 has a long tail of shorter velocities (like from the HDDG measurements) and the modal velocity is about 7 m/s which is closer to the GT velocity.

The PARSIVEL IVDG droplet size distribution (Fig. A4) has a long tail to slower droplets, as can be observed also for the DVSD method in Fig. 6, with a significant number of droplets reported as smaller than the mean. We do not know the cause of these outliers, since the true IVDG droplet size spread is less than 3% (Appendix E). It is possible that stray updrafts in the staircase slow some of the droplets for both sets of measurements.

*4.3. Limitations of experiments*

Our experiments were carried out in a controlled environment using two droplet generators, i.e. HDDG and IVDG. However, unlike real rainfall conditions, there were no strong winds. Moreover, the

drop jets were localized and did not occlude each other. We do not believe that occlusion would be a problem due to the optical arrangement, but the droplet tracks could merge or overlap and the droplets in front or behind the PoF could disturb the measurements. Therefore, it is difficult to predict how well a DVSD would perform under windy conditions or with heavy rainfall.

Our *hourglass* DVSD method (Sect. 2.5.3) requires that the droplets pass through the FoV (see Fig. 1B: left, and Fig. 4: bottom left corner). In an extreme case, the wind could make a droplet trajectory parallel to the line of sight of the camera. In this case, no hourglass would be visible on the DVS recording after an accumulation of events; from the point of view of the DVS, the droplets would appear to shrink and grow while slowly drifting in a random direction.

### 4.4. Future improvements

Improving the stability of our HDDG should be investigated since our HDDG combined with the tiny sampling area of $0.9\,\mathrm{cm}^2$ required patience to capture sufficient good droplets to measure (Appendix D). Using Near Infrared (**NIR**) illumination should also be tried, since most insects would be blind to it and hence not be attracted to the DVSD, and DVS silicon photodiodes work well with NIR illumination. The ring light could also be made much smaller (and less bright) than the large ring that we used.

In principle, it should be possible to infer the 3d trajectory of the droplet from an algorithm that continuously estimates the diameter and velocity of the droplets. We would base this algorithm on cluster trackers commonly used for other DVS applications (Delbruck and Lang 2013; Gallego et al. 2020). The tracker would initiate clusters at the top of the FoV, and then use brightness change events to track the droplets, while measuring their velocity and diameter. A simple set of plausibility checks on the cluster path and cluster size at the start, middle, and end of the path and a fit to the hourglass diameter samples could provide the image plane droplet measurements along with their uncertainties.

The size of the sampling area plays an important role in how quickly a DSD can be obtained. The sampling area of the DVSD decreases slightly with increasing drop size, because the droplets must be fully inside and pass through the FoV. Therefore, a correction will be needed to estimate the DSD to account for the smaller fraction of larger droplets that are measured. This correction would be less important for larger sampling area.

If our DVS were required to have the same sampling area as the OTT Parsivel$^2$ (see Table 4), a reduction in focal length would be necessary, but would increase the current $0.35\,\mathrm{mm}$ drop size uncertainty from 10% (Table 3) to 75%. However, these limitations are a result of the low spatial resolution of our prototype camera. Megapixel DVS are already available (Suh et al. 2020; Finateu et al. 2020). With a megapixel DVS, the $0.35\,\mathrm{mm}$ droplet size uncertainty would be about 20% while matching the $54\,\mathrm{cm}^2$ sampling area of the OTT Parsivel$^2$ (Table 4). Table A1 shows that a 1280x720 pixel DVS with $5\,\mu m$ pixels (Finateu et al. 2020) using a focal length of $50\,\mathrm{mm}$ and a working distance of $75\,\mathrm{mm}$ would provide a sampling area of $48\,\mathrm{cm}^2$ and $0.3\,\mathrm{mm}$ droplets would still occupy 4 pixels. Given the subpixel resolution provided by fitting to the hourglass contour of the accumulated events, the droplet size precision would probably be better than 10%.

### 4.5. Comparing DVSD to other optical disdrometers

Table 4 compares the specifications of our current DVS prototype to the OTT Parsivel$^2$ and 2DVD. Existing optical disdrometers measure the size of the droplets by the size of the 1D occlusion (2DVD) or the decrease in the intensity of light (PARSIVEL). The DVSD takes advantage of the ability of DVS to finely measure the velocity of the droplet across the plane of focus and uses the PoF to locate the droplet in space for unambiguous size measurement.

The measurement uncertainty of the DVSD is in a range similar to that of other optical disdrometers; field experiments with co-located instruments resulted in about 5%–11% diameter error in small drops (D0 = 1–2 mm) and the error varies from approximately 8% to 4.5% at D0 = 1.5 mm from 1 min to 10 min sampling time (Jaffrain and Berne 2012; Chang et al. 2020).

Johannsen et al. (2020) reported difficulty with long-term measurements using PARSIVEL and 2DVD because of drift and insect and spider debris accumulating in optical housings. The simpler free-space optical arrangement of the DVSD could be advantageous in avoiding these problems. Speed is measured by the time of passage between nearby light sheets (2DVD) or by the time that a single light sheet is occluded (PARSIVEL) (Johannsen et al. 2020). Both techniques require high sample rates because droplets that fall at a terminal speed pass through any given point in a few hundred microseconds. *E.g.*, a 1 mm droplet falling at its terminal speed of $4\,\mathrm{m/s}$ (Appendix H) passes by in only $250\,\mu s$. The $1\,\mu s$ time resolution of DVS allows very accurate measurements of droplet speed in the image plane, but at a low camera data rate of 5k to 50k brightness change events per droplet, which could easily be processed by an embedded microcontroller.

**Table 4: Comparison of disdrometer specifications.** Data may not be accurate for latest models.

| Specification | Device | | |
| --- | --- | --- | --- |
| | DVSD[1] | 2DVD[2] | PARSIVEL[3] |
| Technology | 1 dynamic vision sensor | 2 line-scan cameras | Laser-sheet |
| Sensor resolution | $346 \times 260$ | 512 px | 1 photodetector |
| Pixel pitch | $18.5\mu$m | NA | none |
| Power | 3 W camera + 5-40 W LED | 500 W | 1.6 W electronics |
| Data rate | variable (0-1MB/s) | 80 MB/s | 2.4 MB/s |
| Optics | 300mm (HDDG) 75mm (IVDG) | NA | NA |
| Sampling area | 0.88 cm$^2$ (HDDG) 4 cm$^2$ (IVDG) | 100 cm$^2$ | 54 cm$^2$ |
| Diameter range | 0.3–0.6 mm (HDDG) 2.5 mm (IVDG) | 0.1–9.9 mm | 0.2–8.0 mm |
| Speed range | 1–8 m/s | all | 0.2–20.0 m/s |
| Diameter precision | $\pm$0.03 mm (HDDG) | $\pm$0.19 mm | $\pm$2 mm for small |
| Speed precision | $\pm$6% (HDDG) | $\pm$4% | $\pm$5% |

[1] DAVIS346 from www.inivation.com, based on Taverni et al. (2018) FSI sensor chip.
[2] Kruger and Krajewski 2002
[3] *OTT Parsivel$^2$ - Laser Weather Sensor* 2016

## 500  5. Conclusions

Our paper proposes an innovative way to measure droplets using an activity-driven DVS event camera that observes the droplets falling through a shallow DoF. Our results demonstrate the feasibility of this DVSD method for droplets ranging from 0.3 mm to 2.5 mm, covering most of the real rainfall range. Droplet size and velocity measurements from the DVSD have a maximum of 7% MAPE compared to the
ground truth from the drop generator. The droplet size uncertainties of the DVSD measurements and GT values are 10% and 3% respectively, whereas the droplet velocity uncertainties are both 7%. The uncertainty of our prototype is encouraging because we expect substantial potential for improvement from higher resolution event cameras and automated high-throughput processing methods. Our results have smaller bias to overestimate small droplet sizes than the commercial PARSIVEL disdrometer, which
is important for estimating water volume from light rain events.

With our strongest magnifying lens, our DVSD prototype, under laboratory conditions, surpasses SoA disdrometers in terms of precision, although the sampling area is much smaller, as Table 4 shows. For future work, our aim is to increase the sensor resolution and capture real rainfall data with comparisons to SoA disdrometers.

Today, DVSDs would be expensive due to the low volume production costs of DVS cameras. However, mass production for DVS applications in consumer electronics will rapidly decrease production cost and improve the resolution and quality of DVS cameras. The absolute droplet size and velocity depend only on image-plane measurements of the event stream and the values of two stable and easily measured parameters, the magnification $M$ and the camera angle $\alpha$. The use of only a single camera should reduce
cost and requirements for precise alignment. A DVSD based on a low-power, inexpensive embedded Linux microcomputer could be developed that can autonomously estimate droplet diameters and velocities in real time while surviving harsh weather conditions in remote areas disconnected from the power grid. The rain-driven computation and simple optical and lighting requirements of a DVSD would be a great advantage compared to alternative optical disdrometers that sample at a constant high rate and require
more complex optical and lighting arrangements.

## Appendices

## A. Methods and materials

The Matlab code used for data analysis, raw data, simulations, graphs, photos and videos of the experiments is available from the paper website https://sites.google.com/view/dvs-disdrometer/
home.

The appendices consist of the following:

- Appendix C provides details of the Particle Size Velocity (**PARSIVEL**) results presented in App. 3.2.

- Appendix B shows photos of our experimental setups described in App. 3.1.

- Appendix D provides additional useful methodology for testing the stability of the Hard Disk Droplet Generator (**HDDG**) described in App. 2.3.2.

- Appendix E discusses uniformity and long term stability of the Intravenous Dripper Droplet Generator (**IVDG**) from App. 2.3.1.

- Appendix F explains how we optimized our droplet illumination.

- Appendix G explain our calibration of the optical magnification of the camera $M$, and provide tables of computed optical parameters for the experiments and for future Dynamic Vision Sensor Disdrometers (**DVSD**).

- Appendix H explains our model of droplet speed versus fall height and droplet diameter, which we used to obtain our ground truth (**GT**) speed from mass and to ensure that the droplets fell close
to the terminal speed.

## B. Photos of experimental setups

Figs. A1 and A2 show photos of our HDDG and IVDG experimental setups described in App. 2.4.

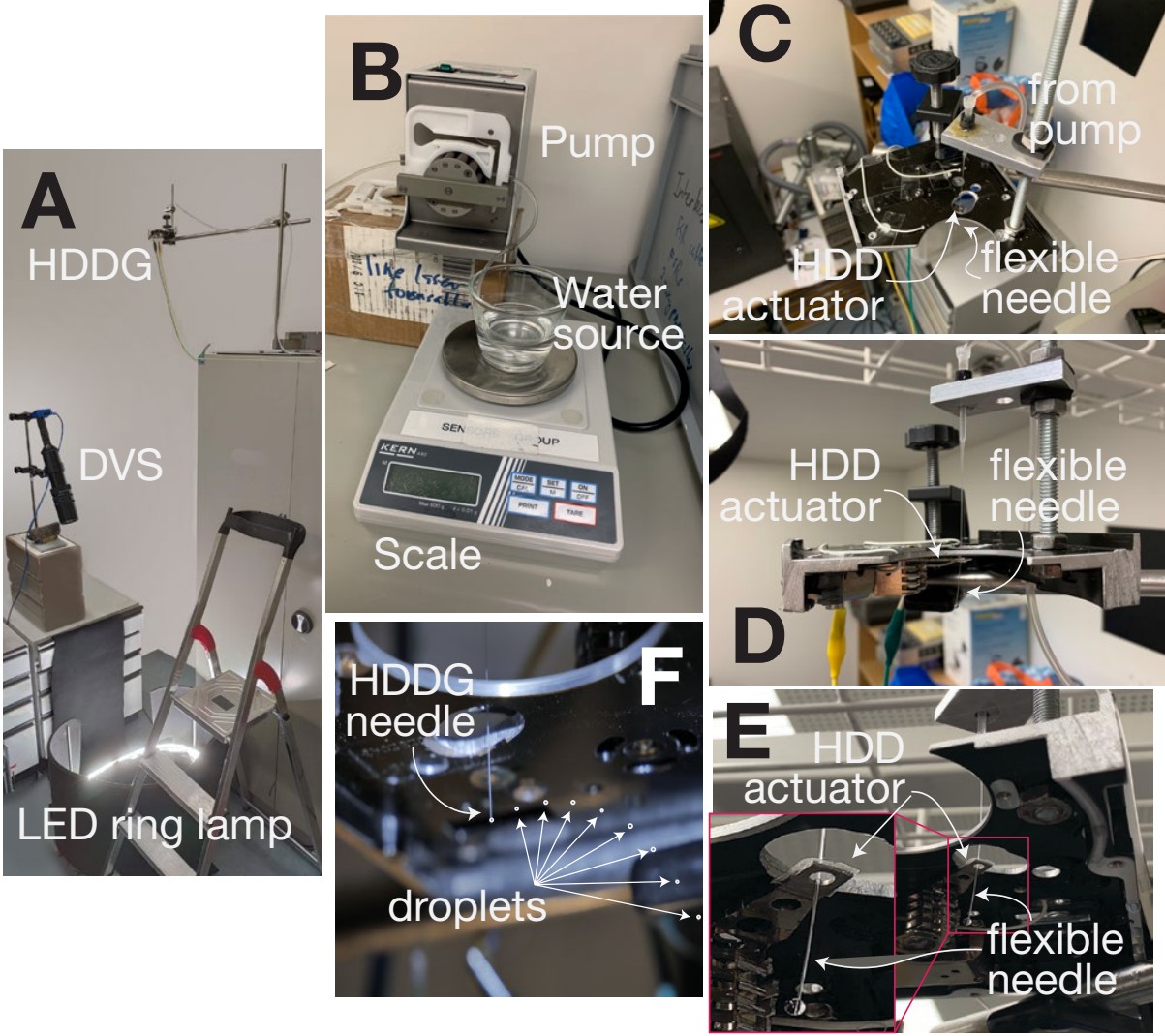

**Figure A1:** Pictures of the HDDG experimental setup. **A**: Overview of the whole HDDG setup. **B**: Peristaltic pump, scale and water tank. **C, D, E**: HDDG drop generator from three different perspectives. **F**: Stream of droplets created by the HDDG, taken with a single short exposure.

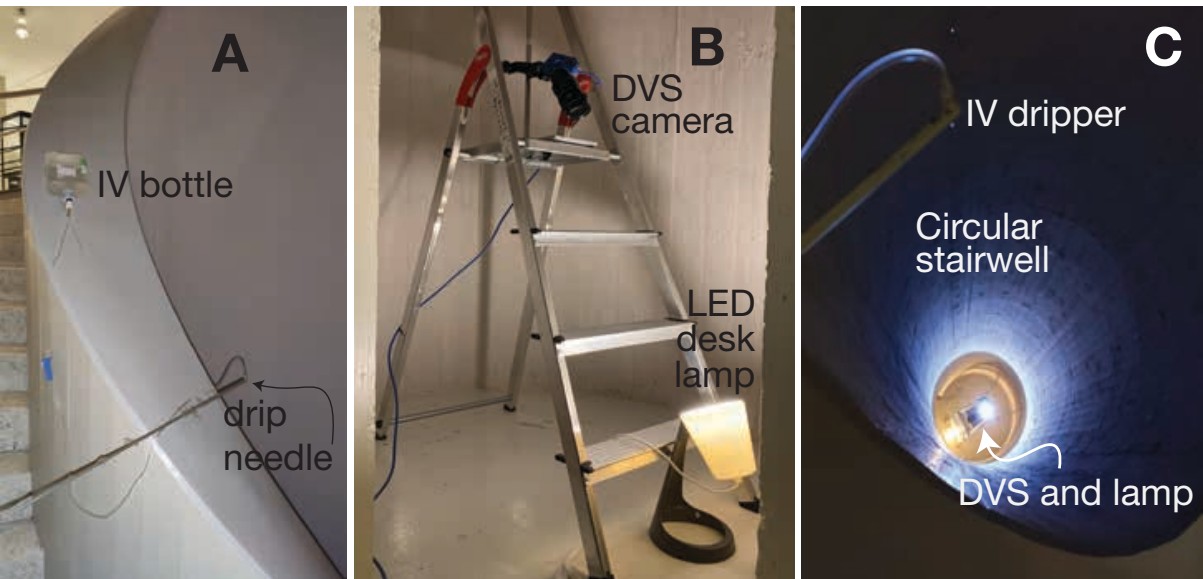

**Figure A2:** Pictures of the IVDG experimental setup. **A**: IVDG drop generator with water source and rod holding the needle. **B**: Dynamic Vision Sensor (**DVS**) camera and lighting setup. **C**: Tube inside the staircase with a fall height of 10 m with the IV dripper at top and DVS camera and lighting setup at the bottom.

## C. Details of **PARSIVEL** results

The sampling area of the PARSIVEL is more than 60X larger than with the DVSD (Table 4) so we could measure hundreds (IVDG) or thousands (HDDG) of droplets under each source droplet condition.

The PARSIVEL HDDG measurements took about 2 m for each droplet size and measured an average of about 3000 droplets. The IVDG measurement took 1200 s to record 149 droplets, an average of one every 8 s, which is about 10X less than the source rate of about 1 Hz. The spatial spread of droplets over the fall of 10 m was still enough to cause many of them miss the 54 cm² PARSIVEL sampling area.

Fig. 7 summarizes the PARSIVEL measurements. Each Fig. A3 plot shows a distribution of measured droplet size and speed for a single GT droplet size, which is marked by an **X** in the plot. The mean of each droplet size and speed measurement is marked by an **O**. The GT droplet sizes were measured using the procedure in App. 2.5.4. Fig. A4 plots the same data as 1D histograms on a linear scale to better see the distribution shapes.

Data collected from the PARSIVEL disdrometer shown in Fig. 7 (see also Appendix C with Fig. A3 and Fig. A4) is binned by the device into droplet size and velocity classes. From the counts of droplets in these classes and the bin size and speed values we computed the mean values shown in Fig. 7.

## D. Testing the stability of the hard disk drop generator with a strobe light

We observed that the HDDG droplet jet continually changed direction a bit. The jet moved in apparent cycles. So we had to be patient and lucky that the jet landed in the DVS measurement area. This is what we mean by "unstable drop generation by HDDG" in Sec. 4. There are probably several effects that caused this behavior, *i.e.*, irregular water flow from the pump, loosely attached needle (so that it can rotate and tilt slightly) and air currents. We believe that air currents only played a minor role and that the most likely culprit is the circular hole in the hard disk drive (HDD) arm that couples it with the needle (Fig. A1E).

Under our lighting and with HDDG drop generation frequencies above 20 Hz, any errors with the HDDG drop release are too fast to be seen with the naked eye. To test the HDDG drop release, we used a custom-built strobe light to illuminate the oscillating needle. For the initial experiments, the goal was to create one drop per cycle, which meant that the drop stream was one-sided. For higher frequencies, it was impossible to tell if a drop is released every one or every two cycles. We built an Arduino-driven LED strobe light to check the drop creation frequency using the principle of aliasing. We set the strobe light frequency to the same as the desired drop creation frequency for two seconds, and for another two seconds, the strobe frequency was half of the desired drop creation frequency. If the distance between successive droplets (see Fig. A1F) changes between the droplets when illuminated with the two different

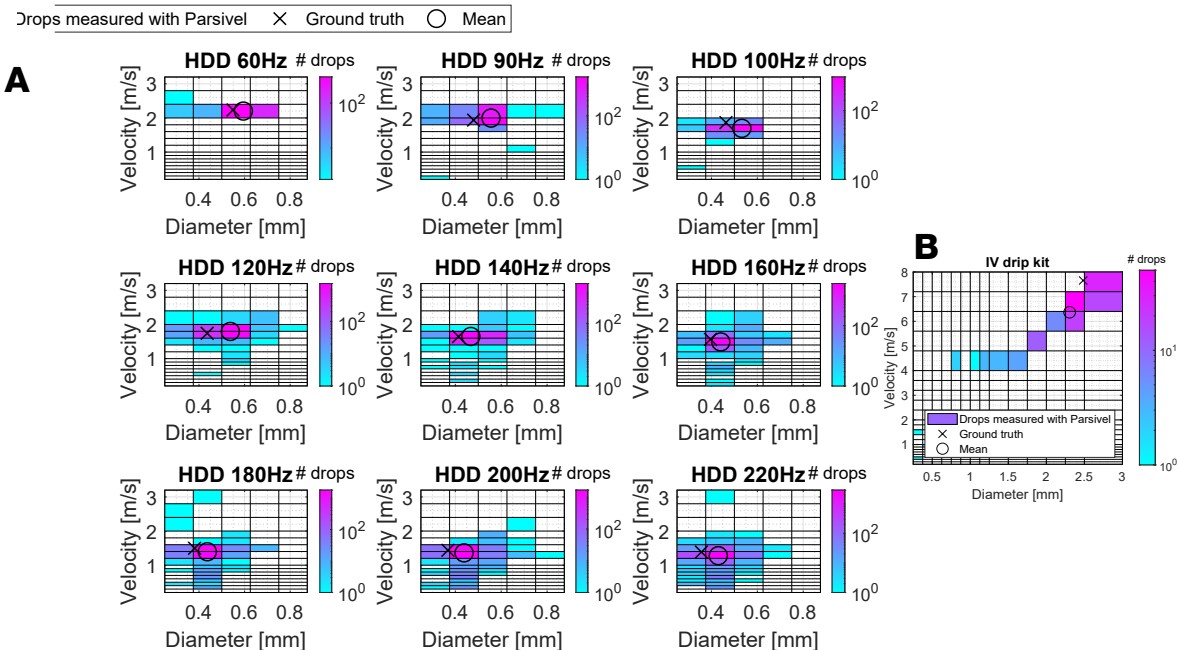

**Figure A3: Results from PARSIVEL in Table 4 plotted as 2D distributions**. The GT droplet size and
speed are indicated by an X in each plot. The means of the distributions are indicated by an O. The bins are
the standard classes produced by the device. **A:** Results from the HDDG. **B:** Results from the IVDG.

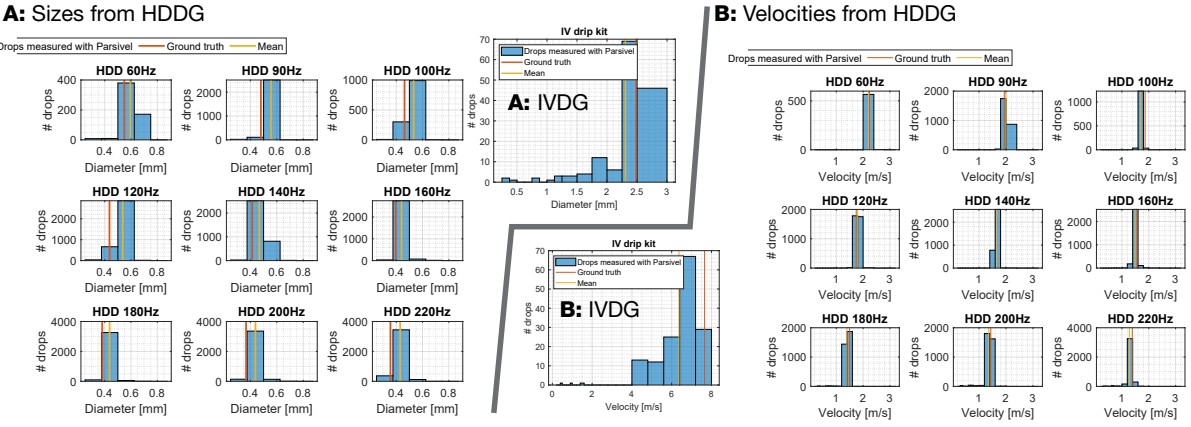

**Figure A4: Results from PARSIVEL in Table 4 plotted as 1D histograms**. The measured ground truth
droplet size and speed are indicated by a vertical red line in each histogram. The mean values are indicated
by the vertical yellow line. The bins are the standard classes produced by the device.**A:** Droplet diameters.
**B:** Droplet velocities.

strobe frequencies (i.e., distance doubles when the strobe light is flashing at half the desired drop creation
frequency), only one drop is released every second cycle. If the distance between the droplets stays the
same, one droplet is released per cycle, which is desired.

## E. Testing uniformity of the IVDG droplets

595 To measure the uniformity of droplets produced by the IVDG (App. 2.3.1), we recorded the droplets
falling from the needle tip with a DVS camera aimed near the tip of the needle. We recorded the event
rate over time and used the distinct peaks in event rate produced by each droplet to measure the droplet
interval with a resolution of 3 ms.

Fig. A5 shows the resulting droplet interval distribution. This measurement took place over 16 minutes and recorded a total of 1200 droplets. During any short time, the droplet dimension variation is less than ±1%. Fig. A3B shows that the droplet interval gradually increased over time. The IVDG measurements in App. 3 were taken over 20 minutes, so they probably had the same slow decrease in droplet size. The total interval increase of 10% over 16m would correspond to a droplet diameter decrease of about 4% over the 20m measurement period of our IVDG experiments.

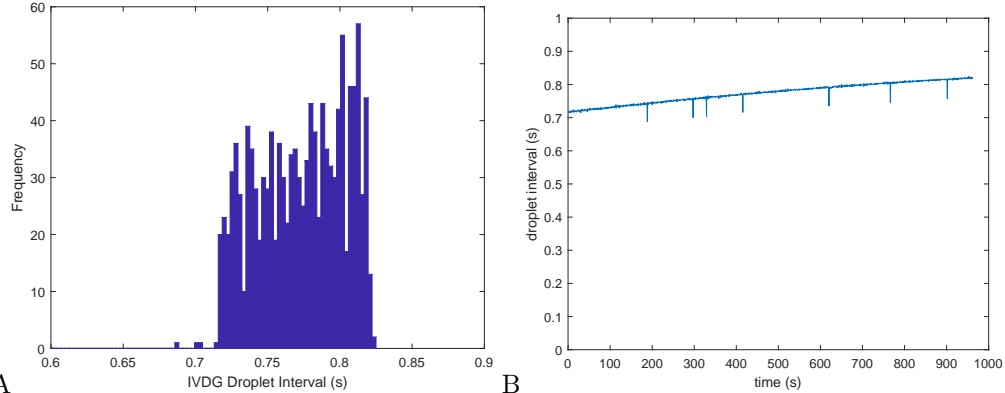

**Figure A5:** IVDG droplet interval distribution (A) and interval vs. time (B).

## F. Optimizing the droplet lighting

Water droplets refract light in the same way that convex glass elements do, namely towards the middle. This property of drops was used to determine a good lighting setup for DVS. To test this optical phenomenon, we took a dispensing needle and attached a tiny water drop at its end and took a photograph of it. We altered the distance between the circular light source and the Plane of Focus (**PoF**) until we were satisfied with the brightness of the drop edge. We note here that the ring light that we used was much too large for our sampling area; we could have used one that was smaller by a factor of about 100 in area. Fig. A6 shows the drop illumination for two different distances of the light source from the droplet. In Fig. A6A the edges of the drop are clearly pronounced, whereas in Fig. A6A the edges are weak. It is important to align the light source correctly as in Fig. A6A, so that the edges of the droplet are well pronounced at the PoF. The light source used for Fig. A6 was a ring light configuration with 6 Light Emitting Diode (**LED**) bulbs. The light source used for the HDDG experiments was a ring light with a LED strip. However, the phenomenon is still the same. For the IVDG setup we were forced to use a single LED desk lamp due to limited stairwell space. This single lamp was not ideal because it caused a bright refraction inside the droplet, but we could still distinguish the outline of the events produced by the droplet edge to measure the droplet diameter.

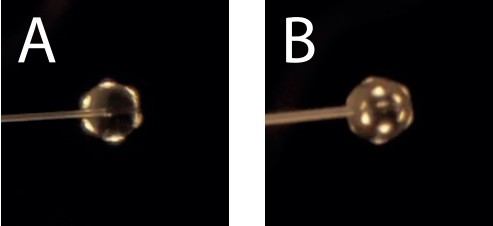

**Figure A6:** Drop illumination from LED ring with 6 lamps, with **A** showing a drop with pronounced edges and **B** showing a drop with less pronounced edges.

## G. Optical and geometrical calibration

The camera extrinsic and intrinsic geometry and optics must be calibrated to compute the physical diameter and velocity measurement from DVS image plane measurements.

*Camera angle $\alpha$.* The parameter $\alpha$ is used in Eq. (2). It is the angle between the Line of Sight (**LoS**) of the camera and the vertical $yz$-plane (see Fig. 1B: left and Fig. A7). We measured $\alpha$ with a smartphone accelerometer.

*Magnification $M$.* The magnification $M$ in px/mm describes how large a certain distance in reality on the PoF would appear on the DVS recording and vice versa. The magnification calibration is done by recording a miniature checkerboard calibration chart held at the PoF with the DVS. $M_{\mathrm{meas}}$ is measured by dividing the number of pixels occupied by a checkerboard square by the checkerboard square size in mm. A comparison of measured $M_{\mathrm{meas}}$ and calculated $M_{\mathrm{calc}}$ values is listed in Table A1 and they are in good agreement. The discrepancy could arise from inaccurate manufacturer focal length specification. We used the measured $M_{\mathrm{meas}}$ value.

### G.1. Field of view, sampling area, and resolution

Fig. A7 shows the geometry of field of view (**FoV**), the Angle of View (**AoV**), which are important for measuring the velocity on the DVS camera, and the sampling area. Table A1 summarizes data of this section. The source spreadsheet is available as an online spreadsheet [a].

The AoV $\theta$ can be calculated with the working distance $w$, horizontal FoV $F_{\mathrm{x}}$ and vertical FoV $F_{\mathrm{y}}$, which are both defined at the PoF. This calculation is done by Eq. (A1):

$$\theta_i = 2\arctan\left(\frac{F_i}{2w}\right) \tag{A1}$$

where $F_i$ is calculated with the measured magnification $M$ in px/mm (Appendix G) and one less than the number of pixels in the according pixel direction of the DVS:

$$\begin{aligned} F_{\mathrm{x}}[\mathrm{mm}] &= \frac{345\ [\mathrm{px}]}{M} \\ F_{\mathrm{y}}[\mathrm{mm}] &= \frac{259\ [\mathrm{px}]}{M} \end{aligned} \tag{A2}$$

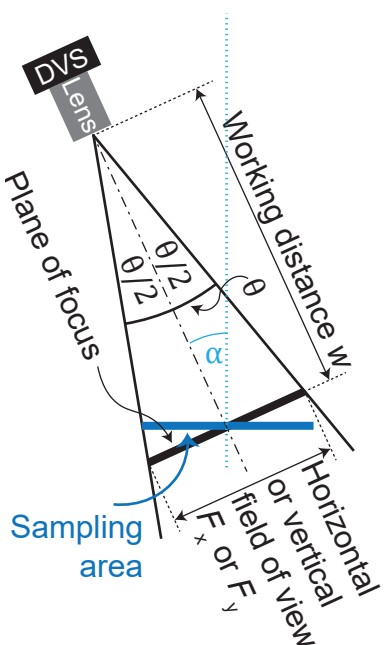

**Figure A7:** Illustration of the FoV $F_{\mathrm{x}}$ and $F_{\mathrm{y}}$ and sampling area of the camera including the corresponding AoV $\theta_{\mathrm{x}}$ and $\theta_{\mathrm{y}}$. The camera angle $\alpha$ is also illustrated.

---

[a]DVS optical parameters spreadsheet

Because the angles $\theta_\mathrm{x}$ and $\theta_\mathrm{y}$ are small, an assumption of a parallel FoV is appropriate ($\theta_i = 0$). The magnification $M$ in the vicinity of the PoF can be assumed to be constant, which simplifies the velocity calculation.

*Sampling area.* The droplet collection sampling area $A_\mathrm{samp}$ in Table 1 is computed from Eq. (A3), which accounts from the camera angle $\alpha$:

$$A_\mathrm{samp} = F_\mathrm{x} F_\mathrm{y} \cos\alpha = \mathrm{SA}_x \mathrm{SA} \tag{A3}$$

*Resolution.* Table A1 lists $\mathrm{px}_{0.3\mathrm{mm}}$ which is the number of pixels occupied by a tiny 0.3mm droplet. Table A1 also lists optical parameters for future DVSDs that use an megapixel DVS with smaller pixels, e.g. Finateu et al. 2020.

*G.2. Depth of Field*

The Depth of Field (**DoF**) is approximately given by Eq. (A4):

$$\mathrm{DoF} = \frac{2w^2 N c}{f^2} \tag{A4}$$

for a given circle of confusion $c$, focal length $f$, $f$-number $N$, and working distance $w$ (Allen and Triantaphillidou 2012). $N$ is the ratio of $f$ to the diameter of the entrance pupil. $c$ is the diameter of the largest image plane spot that is indistinguishable from a point. For this work, we take $c$ to be the DVS 655 pixel size 18.5 um as a means of scaling the arbitrary circle of confusion size to the DVS resolution.

Table A2 summarizes data of this section. The source spreadsheet is available at Footnote *a*. A smaller DoF results in a more pronounced hourglass shape for the accumulated DVS events produced by a droplet. Eq. (A4) shows that we can minimize the DoF by using a fast lens (small $N$) and by maximizing the ratio of focal length to working distance ($f/w$).

| Expt. | format | px size | $n_\mathrm{pix}$ | f | w | $M_\mathrm{calc}$ | $M_\mathrm{meas}$ | $F_x$ | $F_y$ | $\theta_x$ | $\theta_y$ | $\alpha$ | SA | $A_\mathrm{samp}$ | $\mathrm{px}_{0.3\mathrm{mm}}$ |
|---|---|---|---|---|---|---|---|---|---|---|---|---|---|---|---|
| unit | | um | px | mm | cm | px/mm | px/mm | mm | mm | ° | ° | ° | mm×mm | cm$^2$ | px |
| HDDG | 346x260 | 18.5 | 90,567 | 300 | 50 | 32.4 | 30.7 | 11 | 8 | 1.7 | 1.4 | 22 | 11x8 | 0.9 | 9.2 |
| IVDG | 346x260 | 18.5 | 90,567 | 75 | 50 | 8.1 | 7.0 | 49 | 37 | 2.7 | 2.6 | 29.5 | 49x32 | 16.0 | 2.1 |
| Mega1 | 1280x720 | 5 | 923,601 | 50 | 75 | 13.3 | NA | 54 | 96 | 7.3 | 2.8 | 22 | 54x89 | 48.1 | 4.0 |
| Mega2 | 1280x720 | 5 | 923,601 | 50 | 100 | 10.0 | NA | 72 | 128 | 7.3 | 2.8 | 22 | 72x119 | 85.4 | 3.0 |

**Table A1:** HDDG and IVDG geometrical and optical parameters, including possible future megapixel DVS cameras with 5$\mu$m pixels.

| working dist. | f/# | circ. of conf. | focal length | depth of field |
|---|---|---|---|---|
| $w$ | $N$ | $c$ | f | DoF |
| cm | - | um | mm | mm |
| 50 | 4.5 | 18.5 | 300 | 0.5 |
| 50 | 1.2 | 18.5 | 75 | 2.0 |
| 75 | 1.2 | 5 | 50 | 2.7 |
| 100 | 1.2 | 5 | 50 | 4.8 |

**Table A2:** Depth of Field computations, including possible future DVS cameras with 5$\mu$m pixels.

**H. Numerical speed simulation of falling drops**

A numerical velocity simulation was used to analyze the dynamic behavior of a falling droplet with diameters up to 2.5 mm. We used the results of the simulation to determine the terminal speed and the fall height needed to reach any desired final speed, ideally close to the terminal speed. Being close to the terminal speed ensures that DVS captures drops with properties similar to real rainfall, and ensures 665 that an uncertainty in height measurement leads to a small deviation from the predicted velocity by simulation. We determined the GT droplet speeds from the simulation, except for the large IVDG droplets where we used the Gunn and Kinzer (1949) measured speed value.

For simulation, all water drops were assumed to be solid and smooth spheres, which is an eligible approximation especially for drops less than 1 mm that do not experience any significant deformation

according to Beard and Chuang (1987) and Van Boxel et al. (1997). Any effect of deformation or vibration due to air drag and turbulence was neglected. Literature values for the air and water properties were used, where the air and water temperatures for 20 and 25°C were interpolated to 22.5°C.

    The differential equation for the velocity simulation consists of a drag force, gravity, and acceleration term. The differential equation is numerically solved using the Euler forward method. The relation
between drag coefficient and Reynolds number for solid spheres is used, which was fitted to the model of Yang et al. (2015) with the empirically obtained data of Brown and Lawler (2003).

    Numerical velocity simulation is used as the velocity GT to compare velocity measurements with DVS. To determine the accuracy of the simulation, the terminal velocity of the simulation is compared with the accepted reference data of Gunn and Kinzer (1949). The results show that they are very close
to each other for drops up to 1 mm. However, for larger drops, the simulation predicts slightly higher velocities; for the 2.5 mm drops created with the IVDG, the simulation predicts 7.9 m/s whereas Gunn and Kinzer (1949) measured 7.4 m/s.

    Fig. A8A plots the model for different drop diameters. For larger droplets, the fall height must be higher to reach the terminal velocity. Moreover, the terminal velocity for larger drops is larger than for
smaller ones. Fig. A8B compares the simulated velocity and the measured data from Gunn and Kinzer (1949). The simulation starts to differ from the data for droplets with a diameter above 1.5 mm.

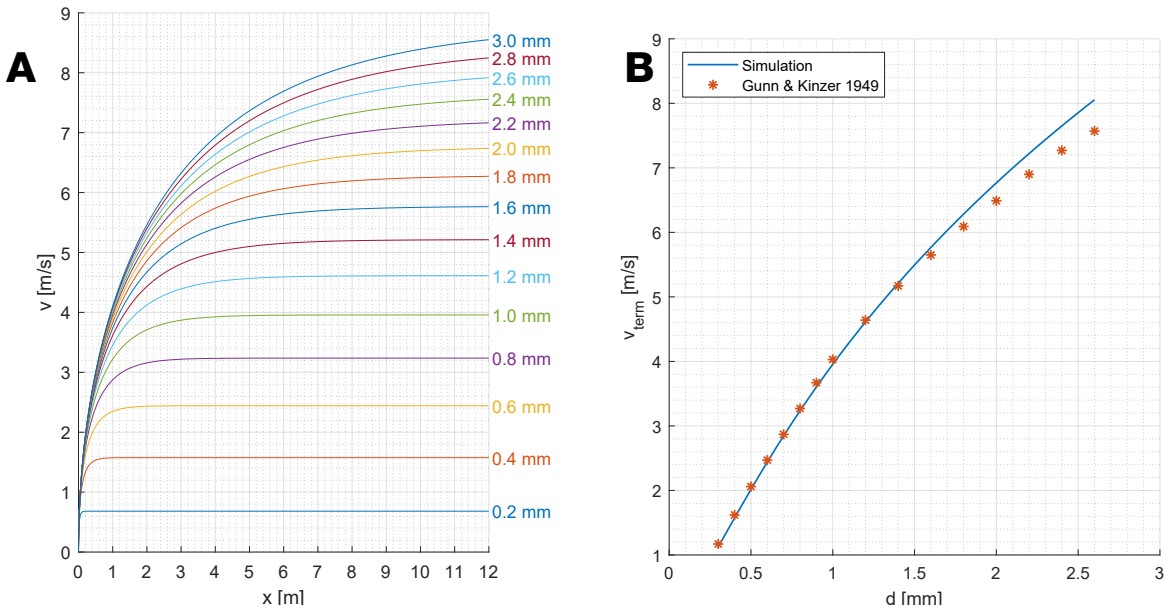

**Figure A8:** Falling droplet simulation results. **A:** Velocity simulation for a fall height up to 12 m and for different drop diameters (written on the right side); vertical displacement $y$ vs. velocity $v$. Literature values for the density of air $\rho_{air}$, kinematic viscosity of air $\nu_{air}$ and density of water $\rho_{water}$ were used for a temperature of 20°C. **B:** Comparison of the terminal simulated terminal velocity to the data of Gunn and Kinzer (1949); diameter $d$ vs. terminal velocity $v_{term}$. Literature values for the density of air $\rho_{air}$, kinematic viscosity of air $\nu_{air}$ and density of water $\rho_{water}$ were used for a temperature of 20°C.

    According to our simulation, a fall height of 2 m is sufficient for drops with a diameter of up to 0.6 mm to reach 99% of the terminal velocity, while for 2.5 mm drops, a fall height of 10 m is necessary to reach 97% of the terminal velocity (see Fig. A8). Thus, a fall height of 2 m was used for the HDDG experiment
and 10 m was used for the IVDG experiment.

### *H.1. Details of droplet speed simulation*

    This following describes the details of the model used to simulate the speed of a falling water droplet. The goal is to find the velocity $v$ of the drop as a function of the vertical distance $y$ the drop has traveled from its initial position for any given diameter $d$. This model allows us to determine the terminal speed
and the required fall height to reach a certain fraction of the terminal speed $v_{term}$, ideally close to the terminal speed. Our model is based on Yang et al. (2015).

The drag force $F_D$ acts on a falling water drop that eventually reaches equilibrium at terminal velocity $v_{term}$ with the gravitational force $mg$. The drag force is defined as Eq. (A1):

$$F_D = \frac{1}{2}\rho_{air}c_D(Re)Av^2 = k(Re)v^2 \tag{A1}$$

where $c_D$ is the drag coefficient depending on the Reynolds number $Re$, $\rho_{air}$ is the density of air, $A$ is the area facing the fluid (for spheres: $A = \pi(\frac{d}{2})^2$) and $k = \frac{1}{2}\rho_{air}c_DA$. The equation of motion can be derived as Eq. (A2):

$$ma = mg - F_D = mg - k(Re)v^2 \tag{A2}$$

where $a$ is the acceleration, $v$ is the velocity, and $g$ is the gravitational acceleration. Eq. (A2) can be rewritten as a differential equation:

$$m\ddot{y} = mg - k(Re)\dot{y}^2 \tag{A3}$$

where $y$ describes the vertical displacement of the droplet from the droplet generator. As mentioned above, the drag coefficient $c_D$ depends on the Reynolds number $Re$. The curve fit for the drag coefficient derived by Yang et al. (2015) is used for the simulation, which is expressed as Eq. (A4):

$$
\begin{aligned}
x &= \frac{\ln(1 + Re)}{10} \\
\alpha &= \left[1 - \exp\left(-3.24x^2 + 8x^4 - 6.5x^5\right)\right] \cdot \frac{\pi}{2} \\
c_D &= \frac{24}{Re} \cdot \left(1 + \frac{3}{16}Re\right)^{0.635} \cdot \cos^3 \alpha + 0.468 \cdot \sin^2 \alpha
\end{aligned}
\tag{A4}
$$

where the Reynolds number $Re$ is defined as

$$Re = \frac{\rho v L}{\mu} = \frac{vL}{\nu}, \tag{A5}$$

where $\rho$ is the density of the fluid, $v$ is the flow velocity, $L$ is the characteristic length (in this case $L = d$), $\mu$ is the dynamic viscosity of the fluid, and $\nu$ is the kinematic viscosity of the fluid. Eq. (A4) is a very good approximation for Reynolds numbers $Re < 2 \times 10^5$, which water drops never exceed, according to Gunn and Kinzer (1949).

MATLAB code to compute discrete time updates of these equations is available from our website (see Appendix H.1).

## Data availability and Supplementary Material

Our Supplementary Material provides our raw data, code, and videos and is available online at the paper web site https://sites.google.com/view/dvs-disdrometer/home.

## Author contributions

JS and KM performed most of the experimental work and data analysis. AA performed initial experiments to establish the concept. TD and JR conceived and supervised the project. All authors contributed to the first draft. TD prepared the final manuscript.

## Competing interests

The authors declare that they have no conflict of interest.

## Acknowledgments

We thank J.P. Carbajal (Ostschweizer Fachhochschule) for catalyzing this collaboration, G. Taverni (formerly UZH-ETH) for help with initial feasibility studies, S. Kosch (aldusleaf.org), N. Ashgriz (U. Toronto), and R. Loidl (UZH-ETH) for their advice on building the HDDG.

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
