# Peer review of "Measuring diameters and velocities of artificial raindrops with a neuromorphic event camera"

_EGUsphere, 2023_

## Author Response (AR1)

**1 Report #1 Submitted on 29 Apr 2023 Anonymous referee #1**

**Reviewer:** The authors present a new technique for measuring size and velocity distributions of artificial raindrops. Overall, I think this is a worthwhile contribution to the field, but the paper needs clarifications in a few places. Further, the paper would benefit from comparisons to other instrumentation for field components of the paper.

**Response**: Thanks very much for this informed and thoughtful review. The major changes to the paper are

1. We measured droplet diameters and velocities using the same HDDG and IVDG droplets using a commercial PARSIVEL disdrometer and report these results in the body as Fig. 10 with error bars and with statistical details in supplement App. A.2. These results are quite consistent with the DVSD results. The PARSIVEL overestimates small droplet volumes by 50% compared with 25% overestimate from the DVSD method.

2. We complete reorganized the paper to a more conventional format with a greatly expanded detailed methodology section taken from our previous supplementary material. The paper now reports the operation of the DVS camera (Sec. 2.2), how the artificial droplets are created (Sec. 2.3), experimental setups (Section 2.4, Figs. 4, A1, A2), the analysis pipeline of droplet selection (Sec. 2.5.1), and the droplet measurement procedures (Secs. 2.5.2 and 2.5.3, Fig. 5). We also report details of our ground truth measurements of source droplet size and speed (Sec. 2.5.4).

Other changes and improvements:

1. Because we have not built an actual disdrometer but only showed the feasibility of using a neuromorphic event camera for such, we re-titled the paper to remove "disdrometer" from the title. Now the title is "Measuring diameters and velocities of artificial raindrops with a neuromorphic event camera".

2. We added detailed spreadsheets of optical parameters and the resulting sampling area (newly illustrated in Fig. A7) and size of 0.3mm droplet as Table A1 and the depth of field computations as Table A2.

3. We measured the uniformity of IVDG droplets and report these measurements as Sec. A.4 and Fig. A5.

4. We modified the GT speed for the large IVDG droplets to be the speed reported in the measurements of Gunn-Kinzer, since these are more accurate than the simulation results, which slightly overestimate the speed.

5. We include expanded discussion at the end of Sec. 4.4 about scaling the DVSD method to the most recently reported megapixel DVS cameras. These scaling computations are detailed in Sec. S.7 and Table A1.

6. We removed the beta parameter from the calibration Sec A.6 because beta is not a calibration parameter – it describes the horizontal component of image plane velocity of each droplet.

**Reviewer:** The authors show small errors in droplet size and velocity. The technique used is simple in its concept, although there are issues that need to be addressed before it is ready to be used by the broader scientific community.

**Response**: We are completely in agreement. Our paper is a feasibility study under laboratory conditions, but we believe that it shows promise for a field DVSD with potential advantages of activity-driven precipitation measurement with simpler optics and potentially lower average power consumption. We retitled the paper to try to indicate the achievements more clearly.

**Reviewer:** I recommend publication after a number of major issues are addressed in light of the comments below.

**Reviewer:** Uncertainty in measurements, size, and velocity in natural/unnatural environments turbulence levels is missing. "In the limitation of experiments, occlusion would be a problem due to the optical arrangement, but the droplet tracks could merge or overlap". What are the upper boundaries of raindrops concentration/turbulence where this technique work?

**Response**: We have not determined these conditions and we believe it goes beyond the scope of the paper. These restrictions are similar, however, to the ones faced by the PARSIVEL which relies on having only a single droplet crossing the laser sheet at one moment. In the DVSD, since each droplet can be tracked across the FoV, we think that it could be quite resistant to occlusion given tracking methods that can make very strong assumptions about the straightness of the droplet paths as they cross the thin plane of focus over a period spanning at most a few milliseconds. Other droplets that cross in front of or behind the plane of focus will create noise or degrade contrast, but they would be so out of focus that the effect might be tolerable. Our paper is already quite long and we hope to leave these important aspects to the field studies of future developments.

**Reviewer:** The authors compare specifications of DVSD, 2DVD, and PARSIVEL, but my biggest concern with the paper is missed opportunity to compare DVSD to other (2DVD and PARSIVEL) common measurements in natural environmental conditions.

**Response**: Thanks. We agree and are happy that these reviews encouraged us to perform PARSIVEL measurements using the same lab setups and report detailed statistics from these measurements by comparison to the DVSD measurements.

**Reviewer:** If I understand correctly, Droplets must pass through the plane of focus Rectangle to measure size. Is there any analysis to estimate the percentage of droplets that missed the target (plane of focus)?

**Response**: We report the effective sampling area in Table 1 and compare it

with other devices in Table 4. The sampling area computations are detailed in Sec. A.7 and Fig A7. The PARSIVEL has a sampling area of 54cm$^2$ and 2DVD has sampling area 80cm$^2$. Our HDD experiments used a tiny sampling area of 0.9cm$^2$ which required a lot of patience to collect sufficient droplets for analysis and our IVDG experiments had a sampling area of only 4cm$^2$. In other words, most HDDG droplets and roughly 9/10 of the IVDG droplets missed these small fingernail sized sampling areas. Detailed computations of the DVSD sampling area for the paper experiments and for possible future DVSD devices that use DVS cameras with higher resolution are reported in Supplemental Table A1, which shows that using a recently published industrial DVS camera, a DVSD with equivalent sampling area as the PARSIVEL would result in 0.3mm droplets with image plane size of 4 pixels. We believe that this could result in competitive small-droplet DSD precision as the PARSIVEL but using only a single camera. This scaling argument is discussed in the paper at the end of Section 4.4 about future improvements.

**Reviewer:** What possible measurements if droplets move in the normal direction to the camera (0)?

**Response**: Section 4.4 discusses this scenario already. In this case, the particle would appear first very out of focus somewhere in the image, then come into focus, and then again go out of focus. An automated event-driven tracking algorithm similar to many that have been previously used with DVS would follow this particle just as if it had appeared at the top and left from the bottom of the image and could still infer its size and speed across the PoF.

**Reviewer:** If you have additional scatter light by small particles, for example, fog particles. Will the measurements affected by it?

**Response**: We can only speculate that small particles in the the thin (mm) plane of focus might cause significant interference. Particles outside the PoF would be out of focus and would mainly reduce the image contrast, but the effect is not known.

**Reviewer:** As you mentioned that droplets must be fully inside of Fov for measurements. If droplets are partially inside the FoV, what is the logic to reject this, and how does this affect the uncertainty?

**Response**: In this study, we manually selected droplets that created hourglass-shaped profiles of accumulated events. Droplets that passed behind or in front of the FoVR (so that they did not create an hourglass) were rejected manually. We believe that this logic properly counts droplets that actually cross the sampling area and can thus be used to infer a correct absolute DSD rate distribution (i.e. rate of droplets of a particular size class). We added this description to the methodology section of the paper in Section 2.5.1 "Data selection". The tiny sampling area of our prototype was only 0.9cm$^2$ which is 60X smaller than the PARSIVEL sampling area means it would take roughly 60X longer to collect the same number of droplets to achieve the same DSD estimation precision. We argue at the end of Sec. 4.4 and show quantitatively in Sec. A.7 that using a megapixel DVS would result in a DVSD with equivalent sampling area as the PARSIVEL with acceptable small-droplet precision – with this device, a 0.3mm

droplet would still occupy 4 pixels in the image plane.

**Reviewer:** Can you explain the logic of different sampling rates for different sizes? And is the sampling rate adjust itself in real time?

**Response**: We regret that this was unclear. The DVS event camera asynchronously reports brightness change events. The sampling rate thus automatically adjusts itself to the necessary rate. We hope that the inclusion of Section 2.2 that describes the operation of the DVS camera helps to clarify this crucial point. The caption of Fig. 1 around line 78 reports our observations that each droplet in our experiments creates from 5k-50k events depending on its image plane size.

**Reviewer:** Minor comments: Line 25-30: The other instruments for particle/rain droplet size distribution in all environmental conditions might be worth highlighting.

**Response**: Thanks for providing these valuable references to these rain and snow disdrometers and their studies of the important effects of wind on measurements. We have added several of them to our introduction and discussion. For brevity we limited our introduction to existing pure optical devices which seem to be an accepted standard for scientific DSD disdrometers. We have no facilities to measure frozen precipitation and thus considered only water in this feasibility study. We also have not done any field measurements with the DVSD method. Unfortunately it would be very difficult to study the effect of wind on measurements in our experimental setups. We note that image-based disdrometers directly measure the size of the hydrometeors with only a one-time calibration of optical magnification and angle of view (M and alpha in our paper). This seems to us to be a potential advantage compared to acoustic and evaporation based devices.

**2 Report #2 Submitted on 29 May 2023 Anonymous referee #2**

**Reviewer:** This paper presents an innovative prototype that aims at measuring size and velocity of falling drops. The topic is relevant for the community and worth publication. However, I believe that some improvements are needed on the current version of the manuscript before publication. I identified three major issues: (i) comparison with measurements obtained with existing devices would greatly strengthen the described results. (ii) I found the paper sometimes hard to follow, while actually find most of the answers in the supplementary material... hence I would suggest to transfer part of the content in the main document. (iii) A description of what occurs when multiple drops fall through the sampling area would be interesting. There is a picture on the supplementary material, but no clear description. Rough estimates of the potential frequency of such event given the sampling volume would also be relevant for the discussion.

**Response**: We are very thankful for your informed review. Together with

considering R1, we have done a major revision to the paper and list the main changes in our first response to reviewer R1.

**Reviewer:** Please find below more detailed comments:

**Reviewer:** - Section 2.1: I found this section rather difficult to read and believe that the content of the A1 (and maybe S3) would be better in the main manuscript to help the reader. The idea of waist of the hourglass shape of the outcome is clear, but I did not understand well how this shape is obtained from the actual output of the DVS. Some clarification would be helpful. Please also clarify how the drop velocity is assessed.

**Response**: This feedback is valuable. We had hoped that our summary Fig. 1 of the methodology would suffice, but clearly it left out necessary details. The paper now includes most of the supplementary material, which explains the complete methodology of the event accumulation to see the hourglass, our manual selection of droplets and how their size and speed are measured. Please check the greatly expanded Sec. 2 of the revised paper and Fig. 5.

**Reviewer:** - Section 2.2: presenting at least briefly the IVDG here would be relevant for reader. Also, adding some details on the uncertainties associated with the size of the generated drops would be interesting (and not only in the supplementary material).

**Response**: Thanks, we agree. We provide details of the IVDG and HDDG in Section 2 of the paper, including photos of the setups in supplemental Figs. A1 and A2.

**Reviewer:** - I would suggest to transfer the content of S4 within the main document since it is very helpful for the reader to grasp how the experiment was actually conducted.

**Response**: Thanks, we have transferred most of the detailed methodology to the body now.

**Reviewer:** - l. 93-94 why only these drop sizes were tested and not also an intermediate one at 1-1.5 mm ?

**Response**: As we explained in the paper, our HDDG could only generate droplets from 0.3 to 0.6mm and our IVDG could only make 2.5mm droplets. Unlike the lovely seminal work of Gunn and Kinzer, we did not have the facilities to supply air to a location with sufficient height to build the air-assisted droplet generator that blows off the droplets when they have not reached the critical mass to break free from surface tension. Our results still show good linearity over the entire range and so we do not believe we could learn much from intermediate droplet size measurements.

**Reviewer:** - Figure 2: this might be a detail but I found the word "Ground truth" quite confusing since it corresponds to the expected size form a mass decrease of the water source in the drop generator which is actually (and obviously !) located at the "top" of the experiment and not on the ground... May

be something like "size of generated drop" would reflect more the experiment.

**Response**: Thanks, we had not made the association with Ground perhaps implying at ground level. The terminology is accepted for other fields. We added a sentence of explanation of this usage at the start of Section 2.5.4. We hope that helps.

**Reviewer:** - l. 163-166: It would be interesting to show how can better results be obtained via corrections to alpha and M estimates.

**Response**: We agree and thank the reviewer for suggesting to consider these effects more carefully. We revised the text in Sec. 4.1 to report our revised understanding of the source of droplet diameter accuracy. For reporting the results, we prefer to use an open-loop method that relies only on the independent measurements of M and alpha and the measured image plane droplet diameter and velocity from hourglass waist width and droplet displacement, since we believe that it honestly reports the basic accuracy of the device without any need for calibration against known droplet sizes and speeds, which could be a great practical advantage for production and field use.

By looking at our (revised) Fig. 9, our DVSD overestimates small droplet diameters by a factor of 0.43/0.4=1.075. Eq. 1 in Sec. 2.6.3 shows that the physical droplet diameter in mm is simply the image diameter in px divided by the camera magnification in px/mm, i.e. $d = d_r/M$. We estimate the image diameter from the accumulated events image (Fig. 7), and it is likely that this procedure slightly overestimates the physical diameter because of pixel quantization. Since the HDDG droplets of .3 to .6 mm measure 9 to 18 px in the image plane, a single pixel over overestimated image plane diameter could explain most of the small droplet diameter discrepancy. We believe an automated algorithm such as discussed in Sec. 4.4 would achieve subpixel accuracy.

Alternatively, it could be that our direct $M_{\mathrm{meas}}$ measurement of 30.7 px/mm (Sec. A.5, Table A1) was slightly off. Table A1 shows that the value of $M_{\mathrm{calc}}$ computed from lens parameters is 32.4, a factor of 1.056X larger. Simply using the computed value of $M_{\mathrm{calc}}$ would bring the HDDG diameter estimates nearly perfectly in alignment with the GT values, but it results in measured droplet velocities that are clearly too small - using the measured M and alpha values results in droplet velocities for small droplets that are nearly perfect. Therefore we believe that the image plane droplet diameter measurements are likely more complex than simply manually measuring the hourglass waist, and could also be complicated by lighting effects as shown in App. A.5 and Fig. A6. Our lighting arrangements have all been preliminary explorations and we have not done any optical modeling to understand the complex refraction of light through the droplets and implications for DVS event streams.

**Reviewer:** - Effect of the wind: the potential effect of wind in real outdoor conditions is briefly mentioned, but I believe that it should be better quantified so that the reader can better understand the real potential of this prototype.

**Response**: Our setup did not allow for any kind of quantified wind generation or measurement, and we believe that although wind and occlusion interference

are clearly important aspects, they lie past the scope of this paper, which is already quite long.

**Reviewer:** - Section 3.1: this section would be much more relevant if it included some comparison with actual measurements from the prototype and the other disdrometers mentioned (or at least one of them).

**Response**: Thanks, we agree and are happy to have been encouraged to compare it with the PARSIVEL device. The methodology, results, and comparison of results are now included in the paper. Detailed statistics of PARSIVEL results are included in the App. A.2.